# From Trees to Continuous Embeddings and Back: Hyperbolic Hierarchical Clustering

Ines Chami[‡]     Albert Gu[†]     Vaggos Chatziafratis[††]     Christopher Ré[†]

[†]Department of Computer Science, Stanford University
[‡]Institute for Computational and Mathematical Engineering, Stanford University
[††]Google Research, NY
{chami, albertgu, vaggos, chrismre}@cs.stanford.edu

## Abstract

Similarity-based Hierarchical Clustering (HC) is a classical unsupervised machine learning algorithm that has traditionally been solved with heuristic algorithms like Average-Linkage. Recently, Dasgupta [25] reframed HC as a discrete optimization problem by introducing a global cost function measuring the quality of a given tree. In this work, we provide the first continuous relaxation of Dasgupta's discrete optimization problem with provable quality guarantees. The key idea of our method, HYPHC, is showing a direct correspondence from discrete trees to continuous representations (via the hyperbolic embeddings of their leaf nodes) and back (via a decoding algorithm that maps leaf embeddings to a dendrogram), allowing us to search the space of discrete binary trees with continuous optimization. Building on analogies between trees and hyperbolic space, we derive a continuous analogue for the notion of lowest common ancestor, which leads to a continuous relaxation of Dasgupta's discrete objective. We can show that after decoding, the global minimizer of our continuous relaxation yields a discrete tree with a $(1 + \varepsilon)$-factor approximation for Dasgupta's optimal tree, where $\varepsilon$ can be made arbitrarily small and controls optimization challenges. We experimentally evaluate HYPHC on a variety of HC benchmarks and find that even approximate solutions found with gradient descent have superior clustering quality than agglomerative heuristics or other gradient based algorithms. Finally, we highlight the flexibility of HYPHC using end-to-end training in a downstream classification task.

## 1   Introduction

Hierarchical Clustering (HC) is a fundamental problem in data analysis, where given datapoints and their pairwise similarities, the goal is to construct a hierarchy over clusters, in the form of a tree whose leaves correspond to datapoints and internal nodes correspond to clusters. HC naturally arises in standard applications where data exhibits hierarchical structure, ranging from phylogenetics [26] and cancer gene sequencing [47, 48] to text/image analysis [49], community detection [36] and everything in between. A family of easy to implement, yet slow, algorithms includes agglomerative methods (e.g., Average-Linkage) that build the tree in a bottom-up manner by iteratively merging pairs of similar datapoints or clusters together. In contrast to "flat" clustering techniques like $k$-means, HC provides fine-grained interpretability and rich cluster information at all levels of granularity and alleviates the requirement of specifying a fixed number of clusters a priori.

Despite the abundance of HC algorithms, the HC theory was underdeveloped, since no "global" objective function was associated with the final tree output. A well-formulated objective allows us to

compare different algorithms, measure their quality, and explain their success or failure.[1] To address this issue, Dasgupta [25] recently introduced a discrete cost function over the space of binary trees with $n$ leaves. A key property of his cost function is that low-cost trees correspond to meaningful hierarchical partitions in the data. He initially proved this for symmetric stochastic block models, and later works provided experimental evidence [44], or showed it for hierarchical stochastic block models, suitable for inputs that contain a ground-truth hierarchical structure [20, 21]. These works led to important steps towards understanding old and building new HC algorithms [14, 15, 40, 4].

The goal of this paper is to improve the performance of HC algorithms using a differentiable relaxation of Dasgupta's discrete optimization problem. There have been recent attempts at gradient-based HC via embedding methods, which do not directly relax Dasgupta's optimization problem. UFit [19] addresses a different "ultrametric fitting" problem using Euclidean embeddings, while gHHC [39] assumes that partial information about the optimal clustering—more specifically leaves' hyperbolic embeddings—is known. Further, these two approaches lack theoretical guarantees in terms of clustering quality and are outperformed by discrete agglomerative algorithms.

A relaxation of Dasgupta's discrete optimization combined with the powerful toolbox of gradient-based optimization has the potential to yield improvements in terms of (a) **clustering quality** (both theoretically and empirically), (b) **scalability**, and (c) **flexibility**, since a gradient-based approach can be integrated into machine learning (ML) pipelines with end-to-end training. However, due to the inherent discreteness of the HC optimization problem, several challenges arise:

(1) How can we continuously **parameterize the search space** of discrete binary trees? A promising direction is leveraging *hyperbolic embeddings* which are more aligned with the geometry of trees than standard Euclidean embeddings [46]. However, hyperbolic embedding methods typically assume a *fully known* [41, 45] or partially known graph [39] that will be embedded, whereas the challenge here is searching over an exponentially large space of trees with *unknown* structure.

(2) How can we derive a **differentiable relaxation** of the HC cost? One of the key challenges is that this cost relies on discrete properties of trees such as the lowest common ancestor (LCA).

(3) How can we **decode** a discrete binary tree from continuous representations, while ensuring that the ultimate discrete cost is close to the continuous relaxation?

Here, we introduce HYPHC, an end-to-end differentiable model for HC with provable guarantees in terms of clustering quality, which can be easily incorporated into ML pipelines.

(1) Rather than minimizing the cost function by optimizing over discrete trees, we parameterize trees using leaves' hyperbolic embeddings. In contrast with Euclidean embeddings, hyperbolic embeddings can represent trees with arbitrarily low distortion in just two dimensions [46]. We show that the leaves themselves provide enough information about the underlying tree, avoiding the need to explicitly represent the discrete structure of internal nodes.

(2) We derive a continuous LCA analogue, which leverages the analogies between shortest paths in trees and hyperbolic geodesics (Fig. 1), and propose a differentiable variant of Dasgupta's cost.

(3) We propose a decoding algorithm for the internal nodes which maps the learned leaf embeddings to a dendrogram (cluster tree) of low distortion.

We show (a) that our approach produces good **clustering quality**, in terms of Dasgupta cost. Theoretically, assuming perfect optimization, the optimal clustering found using HYPHC yields a $(1 + \varepsilon)$-approximation to the minimizer of the discrete cost, where $\varepsilon$ can be made arbitrarily small, and controls the tradeoffs between quality guarantees and optimization challenges. Notice that due to our perfect optimization assumption, this does not contradict previously known computational hardness results based on the Small Set Expansion for achieving constant approximations [14], but allows us to leverage the powerful toolbox of nonconvex optimization. Empirically, we find that even approximate HYPHC solutions found with gradient descent outperform or match the performance of the best discrete and continuous methods on a variety of HC benchmarks. Additionally, (b) we propose two extensions of HYPHC that enable us to **scale** to large inputs and we also study the tradeoffs between clustering quality and speed. Finally, (c) we demonstrate the **flexibility** of HYPHC by jointly optimizing embeddings for HC and a downstream classification task, improving accuracy by up to 3.8% compared to the two-stage embed-then-classify approach.

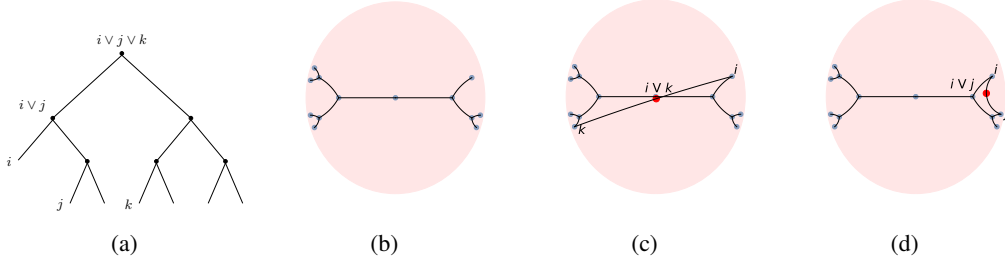

Figure 1: The tree shown in (a) is embedded into hyperbolic space ($\mathbb{B}_2$) in (b), (c), and (d). In (c) and (d) we show the hyperbolic LCA (in red) and illustrate the relationship between the discrete LCA, which is central to Dasgupta's cost, and geodesics (shortest paths) in hyperbolic space.

## 2   Related Work

We review related work for gradient-based HC and refer to Appendix A for an extended discussion of HC related works. Chierchia et al. [19] introduce an ultra-metric fitting framework (UFit) to learn embeddings in ultra-metric spaces. These are restrictive metric spaces (and a special case of tree metrics) where the triangle inequality is strengthened to $d(x, z) \leq \max\{d(x, y), d(y, z)\}$. UFit can be extended to HC using an agglomerative method to produce a dendrogram. gHHC [39], which inspired our work, is the first attempt at HC with hyperbolic embeddings. This model assumes input leaves' hyperbolic embeddings are given (by fixing their distance to the origin) and optimizes embeddings of internal nodes. Observing that the internal tree structure can be directly inferred from leaves' hyperbolic embeddings, we propose an approach that is a direct relaxation of Dasgupta's discrete optimization framework and does not assume any information about the optimal clustering. In contrast with previous gradient-based approaches, our approach has theoretical guarantees in terms of clustering quality and empirically outperforms agglomerative heuristics.

## 3   Background

We first introduce our notations and the problem setup. We then briefly review basic notions from hyperbolic geometry and refer to standard texts for a more in-depth treatment [10]. Appendix B includes additional background about hyperbolic geometry for our technical proofs.

### 3.1   Similarity-Based Hierarchical clustering

**Notations**   We consider a dataset $\mathcal{D}$ with $n$ datapoints and pairwise similarities $(w_{ij})_{i,j\in[n]}$. A HC of $\mathcal{D}$ is a rooted binary tree $T$ with exactly $n$ leaves, such that each leaf corresponds to a datapoint in $\mathcal{D}$, and intermediate nodes represent clusters. For two leaves $(i, j)$ in $T$, we denote $i \vee j$ their LCA, and $T[i \vee j]$ the subtree rooted at $i \vee j$, which represents the smallest cluster containing both $i$ and $j$. In particular, we denote leaves$(T[i \vee j])$ the leaves of $T[i \vee j]$, which correspond to datapoints in the cluster $i \vee j$. Finally, given three leaf nodes $(i, j, k)$ in $T$, we say that the relation $\{i, j | k\}$ holds if $i \vee j$ is a descendant of $i \vee j \vee k$ (Fig. 1a). In this scenario, we have $i \vee k = j \vee k$.

**HC discrete optimization framework**   The goal of HC is to find a binary tree $T$ that is mindful of pairwise similarities in the data. Dasgupta [25] introduced a cost function over possible binary trees with the crucial property that good trees should have a low cost: $C_{\mathrm{Dasgupta}}(T; w) = \sum_{ij} w_{ij} |\mathrm{leaves}(T[i \vee j])|$. Intuitively, a good tree for Dasgupta's cost merges similar nodes (high $w_{ij}$) first in the hierarchy (small subtree). In other words, a good tree should cluster the data such that similar datapoints have LCAs further from the root than dissimilar datapoints. As the quantity $|\mathrm{leaves}(T[i \vee j])|$ may be hard to parse, we note here a simpler way to think of this term based on triplets of datapoints $i, j, k$: whenever a tree splits $i, j, k$ for the first time, then $k \notin \mathrm{leaves}(T[i \vee j])$ if and only if $k$ was separated from $i, j$. In particular, Wang and Wang [50] observe that:

$$C_{\mathrm{Dasgupta}}(T; w) = \sum_{ijk} [w_{ij} + w_{ik} + w_{jk} - w_{ijk}(T; w)] + 2 \sum_{ij} w_{ij}$$

$$\text{where } w_{ijk}(T; w) = w_{ij} \mathbb{1}[\{i, j | k\}] + w_{ik} \mathbb{1}[\{i, k | j\}] + w_{jk} \mathbb{1}[\{j, k | i\}],$$

(1)



Figure 2: Visualization of HYPHC embeddings and decoded trees during optimization.

which reduces difficult-to-compute quantities about subtree sizes to simpler statements about LCAs. For binary trees, exactly one of $\mathbb{1}[\{i, j|k\}], \mathbb{1}[\{i, k|j\}], \mathbb{1}[\{j, k|i\}]$ holds, and these are defined through the notion of LCA. As we shall see next, we can relax this objective using a continuous notion of LCA in hyperbolic space. Dasgupta's cost-based perspective leads to a natural optimization framework for HC, where the goal is to find $T^*$ such that:

$$T^* = \underset{\text{all binary trees } T}{\mathrm{argmin}} \; C_{\text{Dasgupta}}(T; w). \tag{2}$$

### 3.2 Hyperbolic geometry

**Poincaré model of hyperbolic space**  Hyperbolic geometry is a non-Euclidean geometry with a constant negative curvature. We work with the Poincaré model with negative curvature $-1$: $\mathbb{B}_d = \{x \in \mathbb{R}^d : ||x||_2 \le 1\}$. Curvature measures how an object deviates from a flat surface; small absolute curvature values recover Euclidean geometry, while negative curvatures become "tree-like".

**Distance function and geodesics**  The hyperbolic distance between two points $(x, y) \in (\mathbb{B}_d)^2$ is:

$$d(x, y) = \cosh^{-1}\left(1 + 2\frac{||x - y||_2^2}{(1 - ||x||_2^2)(1 - ||y||_2^2)}\right). \tag{3}$$

If $y = o$, the origin of the hyperbolic space, the distance function has the simple expression: $d_o(x) := d(o, x) = 2\tanh^{-1}(||x||_2)$. This distance function induces two types of geodesics: straight lines that go through the origin, and segments of circles perpendicular to the boundary of the ball. These geodesics resemble shortest paths in trees: the hyperbolic distance between two points approaches the sum of distances between the points and the origin, similar to trees, where the shortest path between two nodes goes through their LCA (Fig. 1).

## 4 Hyperbolic Hierarchical Clustering

Our goal is to relax the discrete optimization problem in Eq. (2). To do so, we represent trees using the hyperbolic embeddings of their leaves in the Poincaré disk (Section 4.1).[2] We introduce a differentiable HC cost, using a continuous LCA analogue (Section 4.2), and then propose a decoding algorithm, dec($\cdot$), which maps embeddings to discrete binary trees (Section 4.3). Our main result is that, when embeddings are optimized over a special set of *spread* embeddings (Definition 4.1), solving our continuous relaxation yields a $(1 + \varepsilon)$-approximation to the minimizer of Dasgupta's cost (Section 4.4), where $\varepsilon$ can be made arbitrarily small. Next, we present our continuous optimization framework and then detail the different components of HYPHC.

**HYPHC continuous optimization framework**  Formally, if $\mathcal{Z} \subset \mathbb{B}_2^n$ denotes an arbitrary constrained set of embeddings, we propose to optimize the following continuous constrained optimization problem as a proxy for the discrete problem in Eq. (2):

$$Z^* = \mathrm{argmin}_{Z \in \mathcal{Z}} \; C_{\text{HYPHC}}(Z; w, \tau) \text{ and } T = \mathsf{dec}(Z^*) \tag{4}$$

where $C_{\text{HYPHC}}(\cdot; w, \tau)$ is our differentiable cost and $\tau$ is a temperature used to relax the max function.

## 4.1 Continuous tree representation via hyperbolic embeddings

To perform gradient-based HC, one needs a continuous parameterization of possible binary trees on a fixed number of leaves. We parameterize binary trees using the (learned) hyperbolic embedding of their leaves. Our insight is that because hyperbolic space induces a close correspondence between embeddings and tree metrics [46], the leaf embeddings alone provide enough information to recover the full tree. In fact, as hyperbolic points are pushed towards the boundary, the embedding becomes more tree-like (Fig. 2) [45]. Thus, rather than optimize over all possible tree structures, which could lead to combinatorial explosion, we embed only the leaf nodes in the Poincaré disk using an embedding look-up:

$$i \to z_i \in \mathbb{B}_2. \tag{5}$$

These leaves' embeddings are optimized with gradient-descent using a continuous relaxation of Dasgupta's cost (Section 4.2), and are then decoded into a discrete binary tree to produce a HC on the leaves (Section 4.3).

## 4.2 Differentiable objective function

Having a continuous representation of trees is not sufficient for gradient-based HC, since Dasgupta's cost requires computing the discrete LCA. We leverage the similarities between geodesics in hyperbolic space and shortest paths in trees, to derive a continuous analogue of the discrete LCA. We then use this hyperbolic LCA to introduce a differentiable version of Dasgupta's cost, $C_{\mathrm{HYPHC}}(\cdot; w, \tau)$.

**Hyperbolic lowest common ancestor**   Given two leaf nodes $i$ and $j$, their LCA $i \vee j$ in $T$ is the node on the shortest path connecting $i$ and $j$ (denoted $i \rightsquigarrow j$) that is closest to the root node $r$:

$$i \vee j = \mathrm{argmin}_{k \in i \rightsquigarrow j} d_T(r, k). \tag{6}$$

Analogously, we define the hyperbolic LCA between two points $(z_i, z_j)$ in hyperbolic space as the point on their geodesic (shortest path, denoted $z_i \rightsquigarrow z_j$) that is closest to the origin (the root):

$$z_i \vee z_j := \mathrm{argmin}_{z \in z_i \rightsquigarrow z_j} d(o, z). \tag{7}$$

Note that $z_i \vee z_j$ is also the orthogonal projection of the origin onto the geodesic (Fig. 1) and its hyperbolic distance to the origin can be computed exactly.

**Lemma 4.1.** *Let $(x, y) \in (\mathbb{B}_2)^2$ and $x \vee y$ denote the point on the geodesic connecting $x$ and $y$ that minimizes the distance to the origin $o$. Let $\theta$ be the angle between $(x, y)$ and $\alpha$ be the angle between $(x, x \vee y)$. We have:*

$$\alpha = \tan^{-1}\left(\frac{1}{\sin(\theta)}\left(\frac{||x||_2(||y||_2^2 + 1)}{||y||_2(||x||_2^2 + 1)} - \cos(\theta)\right)\right),$$

$$\text{and } d_o(x \vee y) = 2 \tanh^{-1}(\sqrt{R^2 + 1} - R), \tag{8}$$

$$\text{where } R = \sqrt{\left(\frac{||x||_2^2 + 1}{2||x||_2 \cos(\alpha)}\right)^2 - 1}.$$

We provide a high-level overview of the proof and refer to Appendix C for a detailed derivation. Geodesics in the Poincaré disk are straight lines that go through the origin or segments of circles that are orthogonal to the boundary of the disk. In particular, given two points, one can easily compute the coordinates of the circle that goes through the points and that is orthogonal to the boundary of the disk using circle inversions [9]. One can then use this circle to recover the hyperbolic LCA, which is simply the point on the diameter that is closest to the origin (i.e. smallest Euclidean norm).

**HYPHC's objective function**   The non-differentiable term in Eq. (1) is $w_{ijk}(T; w)$, which is the similarity of the pair that has the deepest LCA in $T$, i.e. the LCA that is farthest—in tree distance—from the root. We use this qualitative interpretation to derive a continuous version of Dasgupta's cost. Consider an embedding of $T$ with $n$ leaves, $Z = \{z_1, \ldots, z_n\}$. The notion of deepest LCA can be extended to continuous embeddings by looking at the continuous LCA that is farthest—in hyperbolic distance—from the origin. Our differentiable HC objective is then:

$$C_{\mathrm{HYPHC}}(Z; w, \tau) = \sum_{ijk}(w_{ij} + w_{ik} + w_{jk} - w_{\mathrm{HYPHC}, ijk}(Z; w, \tau)) + 2\sum_{ij} w_{ij} \tag{9}$$

where $w_{\mathrm{HYPHC}, ijk}(Z; w, \tau) = (w_{ij}, w_{ik}, w_{jk}) \cdot \sigma_\tau(d_o(z_i \vee z_j), d_o(z_i \vee z_k), d_o(z_j \vee z_k))^\top$,

and $\sigma_\tau(\cdot)$ is the scaled softmax function: $\sigma_\tau(\alpha)_i = e^{\alpha_i/\tau}/\sum_j e^{\alpha_j/\tau}$.

## 4.3 Hyperbolic decoding

The output of HC needs to be a discrete binary tree, and optimizing the HYPHC loss in Eq. (4) only produces leaves' embeddings. We propose a way to decode a binary tree structure from embeddings by iteratively merging the most similar pairs based on their hyperbolic LCA distance to the origin (Algorithm 1). Intuitively, because $\mathrm{dec}(\cdot)$ uses LCA distances to the origin (and not pairwise distances), it can recover the underlying tree that is directly being optimized by HYPHC (which is only defined through these LCA depths).

---

**Algorithm 1** Hyperbolic binary tree decoding $\mathrm{dec}(Z)$

---

1: **Input:** Embeddings $Z = \{z_1, \ldots, z_n\}$; **Output:** Rooted binary tree with $n$ leaves.
2: $F \leftarrow$ forest $(\{i\} : i \in [n])$
3: $S \leftarrow \{(i,j)$: pairs sorted by deepest hyperbolic LCA $(d_o(z_i \vee z_j))\}$;
4: **for** $(i,j) \in S$ **do**
5:      **if** $i$ and $j$ not in same tree in $F$ **then**
6:          $r_i, r_j \leftarrow$ roots of trees containing $i, j$;
7:          create new node with children $r_i, r_j$;
8: **return** $F$ (which is a binary tree at the end of the algorithm)

---

## 4.4 Approximation ratio result

Assuming one can perfectly solve the constrained optimization in Eq. (4), our main result is that the solution of the continuous optimization, once decoded, recovers the optimal tree with a constant approximation factor, which can be made arbitrarily small. We first provide more conditions on the constrained set of embeddings in Eq. (4) by introducing a special set of *spread* embeddings $\mathcal{Z} \subset \mathbb{B}_2^n$:

**Definition 4.1.** *An embedding $Z \in \mathbb{B}_2^n$ is called* spread *if for every triplet $(i, j, k)$:*

$$\max\{d_o(z_i \vee z_j), d_o(z_i \vee z_k), d_o(z_j \vee z_k)\} - \min\{d_o(z_i \vee z_j), d_o(z_i \vee z_k), d_o(z_j \vee z_k)\} > \delta \cdot O(n), \tag{10}$$

*where $\delta$ is a constant of hyperbolic space (Gromov's delta hyperbolicity, see Appendix B.2).*

Intuitively, the spread constraints spread points apart and force hyperbolic LCAs to be distinguishable from each other. Theoretically, we show that this induces a direct correspondence between embeddings and binary trees, i.e. the embeddings yield a metric that is close to that of a binary tree metric. More concretely, every spread leaf embedding is compatible with an underlying tree, in the sense that our decoding algorithm is guaranteed to return the underlying tree. This is the essence of the correspondence between spread embeddings and trees, which is the main ingredient behind Theorem 4.1.

**Theorem 4.1.** *Consider a dataset with $n$ datapoints and pairwise similarities $(w_{ij})$ and let $T^*$ be the solution of Eq. (2). Let $\mathcal{Z}$ be the set of spread embeddings and $Z^* \in \mathcal{Z}$ be the solution of Eq. (4) for some $\tau > 0$. For any $\varepsilon > 0$, if $\tau \leq \mathcal{O}(1/\log(1/\varepsilon))$, then:*

$$\frac{C_{\mathrm{Dasgupta}}(\mathrm{dec}(Z^*); w)}{C_{\mathrm{Dasgupta}}(T^*; w)} \leq 1 + \varepsilon. \tag{11}$$

The proof is detailed in Appendix D. The insight is that when embeddings are sufficiently spread out, there is an equivalence between leaf embeddings and trees. In one direction, we show that *any* spread embedding $Z$ has a continuous cost $C_{\mathrm{HYPHC}}(Z; w, \tau)$ close to the discrete cost $C_{\mathrm{Dasgupta}}(T; w)$ of some underlying tree $T$. Conversely, we can leverage classical hyperbolic embedding results [46] to show that Dasgupta's optimum $T^*$ can be embedded as a spread embedding in $\mathcal{Z}$.

**Optimization challenges** Note that smaller $\tau$ values in Eq. (11) lead to better approximation guarantees but make the optimization more difficult. Indeed, hardness in the discrete problem arises from the difficulty in enumerating all possible solutions with discrete exhaustive search; on the other hand, in this continuous setting it is easier to specify the solution, which trades off for challenges arising from nonconvex optimization—i.e. how do we find the global minimizer of the constrained optimization problem in Eq. (4). In our experiments, we use a small constant for $\tau$ following standard

|  |  | Zoo | Iris | Glass | Segmentation | Spambase | Letter | CIFAR-100 |
|---|---|---|---|---|---|---|---|---|
|  | # Points | 101 | 150 | 214 | 2310 | 4601 | 20K | 50K |
|  | Cost | $DC.10^{-5}$ | $DC.10^{-5}$ | $DC.10^{-6}$ | $DC.10^{-9}$ | $DC.10^{-10}$ | $DC.10^{-12}$ | $DC.10^{-13}$ |
|  | Upper Bound | 3.887 | 14.12 | 3.959 | 4.839 | 3.495 | 3.031 | 4.408 |
|  | Lower Bound | 2.750 | 7.709 | 2.750 | 3.258 | 3.025 | 2.244 | 3.990 |
| Discrete | SL | 2.897 | 8.120 | 3.018 | 3.705 | 3.250 | 2.625 | 4.149 |
|  | AL | 2.829 | 7.939 | 2.906 | 3.408 | 3.159 | 2.437 | 4.056 |
|  | CL | **2.802** | 7.950 | 2.939 | 3.460 | 3.184 | 2.481 | 4.078 |
|  | WL | 2.827 | 7.938 | 2.920 | 3.434 | 3.170 | 2.453 | 4.060 |
|  | BKM | 2.861 | 8.223 | 2.948 | 3.375 | 3.127 | **2.383** | 4.056 |
| Continuous | UFit | 2.896 | 7.916 | 2.925 | 3.560 | 3.204 | - | - |
|  | HYPHC | **2.802** | **7.881** | **2.902** | **3.341** | **3.126** | 2.384 | **4.053** |

Table 1: Clustering quality results measured in discrete Dasgupta's cost (DC). Best score in bold, second best underlined. Dashes indicate that the method could not scale to large datasets.

practice. To optimize over the set of spread embeddings, one could add triplet constraints on the embeddings to the objective function. We empirically find that even solutions of the *unconstrained* problem found with stochastic gradient descent have a good approximation ratio for the discrete objective, compared to baseline methods (Section 6.2).

## 5   HYPHC Practical Considerations

Here, we propose two empirical techniques to reduce the runtime complexity of HYPHC and improve its scalability, namely triplet sampling and a greedy decoding algorithm. Note that while our theoretical guarantees in Theorem 4.1 are valid under the assumptions that we use the exact decoding and compute the exact loss, we empirically validate that HYPHC, combined with these two techniques, still produces a clustering of good quality in terms of Dasgupta's cost (Section 6.3).

**Greedy decoding**   Algorithm 1 requires solving the closest pair problem, for which an almost-quadratic $n^{2-o(1)}$ lower bound holds under plausible complexity-theoretic assumptions [3, 22]. Instead, we propose a greedy top-down decoding that runs significantly faster. We normalize leaves' embeddings so that they lie on a hyperbolic diameter and sort them based on their angles in the Poincaré disk. We then recursively split the data into subsets using the largest angle split (Fig. 8 in Appendix E.2). Intuitively, if points are normalized, the LCA distance to the origin is a monotonic function of the angle between points. Therefore, this top-down decoding acts as a proxy for the bottom-up decoding in Algorithm 1. In the worst case, this algorithm runs in quadratic time. However, if the repeated data splits are roughly evenly sized, it can be asymptotically faster, i.e., $O(n \log n)$. In our experiments, we observe that this algorithm is significantly faster than the exact decoding.

**Triplet sampling**   Computing the loss term in Eq. (9) requires summing over all triplets which takes $O(n^3)$ time. Instead, we generate all unique pairs of leaves and sample a third leaf uniformly at random from all the other leaves, which yields roughly $n^2$ triplets. Note that an important benefit of HYPHC is that no matter how big the input is, HYPHC can always produce a clustering by sampling fewer triplets. We view this as an interesting opportunity to scale to large datasets and discuss the scalability-clustering quality tradeoff of HYPHC in Section 6.3. Finally, note that triplet sampling can be made parallel, unlike agglomerative algorithms, even if both take $\mathcal{O}(n^2)$ time.

## 6   Experiments

We first describe our experimental protocol (Section 6.1) and evaluate the **clustering quality** of HYPHC in terms of discrete Dasgupta cost (Section 6.2). We then analyze the clustering **quality/speed tradeoffs** of HYPHC (Section 6.3). Finally, we demonstrate the **flexibility** of HYPHC using end-to-end training for a downstream classification task (Section 6.4).

### 6.1   Experimental setup

We describe our experimental setup and refer to Appendix E.1 for more details.

**Datasets**  We measure the clustering quality of HYPHC on six standard datasets from the UCI Machine Learning repository,[3] as well as CIFAR-100 [35], which exhibits a hierarchical structure (each image belongs to a fine-grained class that is itself part of a coarse superclass). Note that the setting studied in this work is similarity-based HC, where the input is only pairwise similarities, rather than features representing the datapoints. For all datasets, we use the cosine similarity to compute a complete input similarity graph.

**Baselines**  We compare HYPHC to similarity-based HC methods, including competitive agglomerative clustering approaches such as single, average, complete and Ward Linkage (SL, AL, CL and WL respectively). We also compare to Bisecting K-Means (BKM) [40], which is a fast top-down algorithm that splits the data into two clusters at every iteration using local search.[4]  Finally, we compare to the recent gradient-based Ultrametric Fitting (UFit) approach [19].[5]

**Evaluation metrics**  Our goal in this work is not to show an advantage on different heuristics, but rather to optimize a single well-defined search problem to the best of our abilities. We therefore measure the clustering quality by computing the discrete Dasgupta Cost (DC). A lower DC means a better clustering quality. We also report upper and lower bounds for DC (defined in the Appendix E.1). For classification experiments (Section 6.4) where the goal is to predict labels, we measure the classification accuracy.

**Training procedure**  We train HYPHC for 50 epochs (of the sampled triples) and optimize embeddings with Riemannian Adam [7]. We set the embedding dimension to two in all experiments, and normalize embeddings during optimization as described in the greedy decoding strategy (Section 5). We perform a hyper-parameter search over learning rate values $[1e^{-3}, 5e^{-4}, 1e^{-4}]$ and temperature values $[1e^{-1}, 5e^{-2}, 1e^{-2}]$.

**Implementation**  We implemented HypHC in PyTorch and make our implementation publicly available.[6] To optimize the HYPHC loss in Eq. (9), we used the open-source Riemannian optimization software geoopt [32]. We conducted our experiments on a single NVIDIA Tesla P100 GPU.

## 6.2 Clustering quality

We report the performance of HYPHC—fast version with greedy decoding and triplet sampling—in Table 1, and compare to baseline methods. On all datasets, HYPHC outperforms or matches the performance of the best discrete method, and significantly improves over UFit, the only similarity-based continuous method. This confirms our intuition that directly optimizing a continuous relaxation of Dasgupta's objective can improve clustering quality, compared to baselines that are optimized independently of the objective.

We visualize embeddings during different iterations of HYPHC on the zoo dataset in Fig. 2. Colors indicate ground truth flat clusters for the datapoints and we observe that these are better separated in the dendrogram as optimization progresses. We note that embeddings are pushed towards the boundary of the disk, where the hyperbolic distances are more "tree-like" and produce a better HC. This illustrates our intuition that the optimal embedding will be close to a tree metric embedding.

## 6.3 Analysis

In our experiments, we used the greedy decoding Algorithm and triplet sampling with $\mathcal{O}(n^2)$ triplets (Section 5). Since our theoretical guarantees apply to the full triplet loss and exact decoding, we are interested in understanding how much quality is lost by using these two empirical techniques.

**Decoding**  We report HC costs obtained using the exact (Section 4.3) and the greedy (Section 5) decoding in Fig. 3b, as well as the corresponding runtime in milliseconds (ms), averaged over 10 runs. We observe that the greedy decoding is approximately 60 times faster than the exact decoding, while still achieving almost the same cost. This confirms that, when embeddings are normalized in two dimensions, using angles as a proxy for the LCA's distances to the origin is a valid approach.

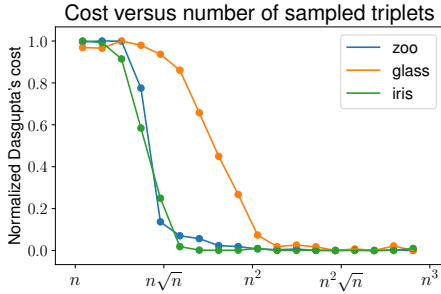

(a) Normalized cost vs number of triplets.

| | | Zoo | Iris | Glass | Segmentation |
|---|---|---|---|---|---|
| Cost | Exact | $\mathbf{2.8016}.10^{-5}$ | $\mathbf{7.8783}.10^{-5}$ | $\mathbf{2.9011}.10^{-6}$ | $\mathbf{3.3417}.10^{-9}$ |
| | Greedy | $2.8021.10^{-5}$ | $7.8807.10^{-5}$ | $2.9019.10^{-6}$ | $3.3419.10^{-9}$ |
| Time (ms) | Exact | 78 | 323 | 400 | 21717 |
| | Greedy | 3.1 | 5.2 | 6.2 | 303 |
| | Speedup | 25x | 62x | 64x | 72x |

(b) Speed/quality analysis for the exact and greedy decoding.

| | Zoo | Iris | Glass | Segmentation |
|---|---|---|---|---|
| # Classes | 7 | 3 | 6 | 7 |
| LP | 41.4 | 76.7 | 46.8 | 65.3 |
| HYPHC-Two-Step | $84.8 \pm 3.5$ | $84.4 \pm 1.7$ | $50.6 \pm 2.6$ | $64.1 \pm 0.9$ |
| HYPHC-End-to-End | $\mathbf{87.9 \pm 3.8}$ | $\mathbf{85.6 \pm 0.8}$ | $\mathbf{54.4 \pm 2.9}$ | $\mathbf{67.7 \pm 3.4}$ |

(c) Classification accuracy for different training strategies.

Figure 3: (a): Triplet sampling analysis. (b): Downstream classification task. (c): Decoding analysis.

**Triplet sampling** We plot the discrete cost (normalized to be in $[0, 1]$) of HYPHC for different number of sampled triplets in Fig. 3a. As expected, we note that increasing the number of sampled triplets reduces DC, and therefore improves the clustering quality. This tradeoff forms one of the key benefits of using gradient-based methods: HYPHC can always produce a hierarchical clustering, no matter how large the input is, at the cost of reducing clustering quality or increasing runtime. In our experiments, we find that $O(n^2)$ is sufficient to achieve good clustering quality, which is the most expensive step of HYPHC in terms of runtime complexity. Future work could explore better triplet sampling strategies, to potentially use less triplets while still achieving a good clustering quality.

## 6.4 End-to-end training

Here, we demonstrate the flexibility of our approach and the benefits of joint training on a downstream similarity-based classification task. Since HYPHC is optimized with gradient-descent, it can be used in conjunction with any standard ML pipeline, such as downstream classification. We consider four of the HC datasets that come with categorical labels for leaf nodes, split into training, testing and validation sets (30/60/10% splits). We follow a standard graph-based semi-supervised learning setting, where all the nodes (trainining/validation/testing) are available at training time, but only training labels can be used to train the models. We use the embeddings learned by HYPHC as input features for a hyperbolic logistic regression model [27]. Note that none of the other discrete HC methods apply here, since these do not produce representations. In Fig. 3c, we compare jointly optimizing the HYPHC and the classification loss (End-to-End) versus a two-step embed-then-classify approach which trains the classification module using frozen HYPHC embeddings (Two-Step) (average scores and standard deviation computed over 5 runs). We also compare to Label Propagation (LP), which is a simple graph-based semi-supervised algorithm that does not perform any clustering step. We find that LP is outperformed by both the end-to-end and the two-step approaches on most datasets, suggesting that clustering learns meaningful partitions of the input similarity graph. Further, we observe that end-to-end training improves classification accuracy by up to 3.8%, confirming the benefits of a differentiable method for HC.

## 7 Conclusion

We introduced HYPHC, a differentiable approach to learn HC with gradient-descent in hyperbolic space. HYPHC uses a novel technical approach to optimize over discrete trees, by showing an equivalence between trees and constrained hyperbolic embeddings. Theoretically, HYPHC has a $(1 + \varepsilon)$-factor approximation to the minimizer of Dasgupta's cost, and empirically, HYPHC outperforms existing HC algorithms. While our theoretical analysis assumes a perfect optimization, interesting future directions include a better characterization of the hardness arising from optimization challenges, as well as providing an approximation for the continuous optimum. While no constant factor approximation of the continuous optimum is possible, achieving better (e.g. polylogarithmic) approximations, is an interesting future direction for this work. Finally, we note that our continuous optimization framework can be extended beyond the scope of HC, to applications that require searching of discrete tree structures, such as constituency parsing in natural language processing.

## Broader Impact

Clustering is arguably one of the most commonly used tools in computer science applications. Here, we study a variation where the goal is to output a hierarchy over clusters, as data often contain hierarchical structures. We believe our approach based on triplet sampling and optimization should not raise any ethical considerations, to the extent that the input data for our algorithm is unbiased. Of course, bias in data is by itself another challenging problem, as biases can lead to unfair clustering and discriminatory decisions for different datapoints. However here we study a downstream application, *after* data has been collected. As such, we hope only a positive impact can emerge from our work, by more faithfully finding hierarchies in biological, financial, or network data, as these are only some of the applications that we listed in the introduction.

## Acknowledgments

We gratefully acknowledge the support of DARPA under Nos. FA86501827865 (SDH) and FA86501827882 (ASED); NIH under No. U54EB020405 (Mobilize), NSF under Nos. CCF1763315 (Beyond Sparsity), CCF1563078 (Volume to Velocity), and 1937301 (RTML); ONR under No. N000141712266 (Unifying Weak Supervision); the Moore Foundation, NXP, Xilinx, LETI-CEA, Intel, IBM, Microsoft, NEC, Toshiba, TSMC, ARM, Hitachi, BASF, Accenture, Ericsson, Qualcomm, Analog Devices, the Okawa Foundation, American Family Insurance, Google Cloud, Swiss Re, the HAI-AWS Cloud Credits for Research program, TOTAL, and members of the Stanford DAWN project: Teradata, Facebook, Google, Ant Financial, NEC, VMWare, and Infosys. The U.S. Government is authorized to reproduce and distribute reprints for Governmental purposes notwithstanding any copyright notation thereon. Any opinions, findings, and conclusions or recommendations expressed in this material are those of the authors and do not necessarily reflect the views, policies, or endorsements, either expressed or implied, of DARPA, NIH, ONR, or the U.S. Government.

## Footnotes

[1]Contrast this lack of HC objectives with flat clustering, where $k$-means objectives and others have been studied intensively from as early as 1960s (e.g., [29]), leading today to a comprehensive theory on clustering.

[2]We developed our theory in two dimensions. As shown in Sala et al. [45], this can present optimization difficulties due to precision requirements. However, using Prop 3.1 of [45], we can increase the dimension to reduce this effect.

[3] https://archive.ics.uci.edu/ml/datasets.php

[4] BKM is the direct analog of Hierarchical K-Means in the context of similarity-based HC [40].

[5] Note that we do not directly compare to gHHC [39] since this method requires input features. For completeness, we include a comparison in the Appendix E.3.

[6] https://github.com/HazyResearch/HypHC

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
