[Supplementary Material · HypHC_supplemental.pdf]

# A  Further Related Work

While we have covered the most relevant related work and background in the main body, we give more details for related HC methods and objectives.

**Dasgupta's cost**  Having an objective function for HC is crucial not only for guiding the optimization, but also for having theoretically grounded solutions. Unfortunately, there has been a lack of global objectives measuring the quality of a HC, which is in stark contrast with the plethora of flat clustering objectives like $k$-means, $k$-center, correlation clustering (e.g., [17, 5]) and many more. This lack of optimization objectives for HC was initially addressed indirectly in [24] (by comparing a tree against solutions to $k$-clustering objectives for multiple $k$), and has again emerged recently by Dasgupta [25], who introduced a cost to evaluate and compare the performance of HC algorithms. This discrete HC cost has the key property that good clustering trees should yield a low cost. The formulation of this objective favors binary trees and the optimum solution can always be assumed to be binary, as a tree with higher fan-out in any internal node can easily be modified into a binary tree with the same cost (or less). In particular, one interesting aspect of this objective is that running Recursive Sparsest Cut would produce a HC with provable guarantees with respect to his cost function [14]. Subsequent work was able to shed light to Average Linkage performance; specifically, [40] studied the complement to Dasgupta's cost function and showed that Average Linkage will find a solution within 33% of the optimizer. Further techniques based on semidefinite programming relaxations led to improved approximations [15, 2, 4], however with a significant overhead in runtime (see [18] for a survey).

**Agglomerative Hierachical Clustering**  One of the first suite of algorithms developed to solve HC was bottom-up linkage methods. These are simple and easy to implement algorithms that recursively merge similar datapoints to form some small clusters and then gradually larger and larger clusters emerge. Well-known heuristics include Single Linkage, Complete Linkage and Average Linkage, that we briefly describe here. All three heuristics start with $n$ datapoints forming singleton clusters initially and perform exactly $n - 1$ merges in total until they produce a binary tree corresponding to the HC output. At any given step, if $A$ and $B$ denote two already formed clusters, the criterion for which clusters to merge next is to *minimize* the minimum, maximum and average distance between two clusters $A$ and $B$, for Single, Complete and Average Linkage respectively. If instead of pairwise distances the input was given as a similarity graph, analogous formulas can be used, where the criterion is to maximize the respective quantities. These algorithms can be made to run in time $O(n^2 \log n)$. A bottleneck at the core of the computations for such algorithms is the nearest neighbor problem, for which known almost quadratic lower bounds apply [22, 3]. However, recent works [1, 16] try to address this in Euclidean spaces where we are provided features by using locality sensitive hashing techniques and approximate nearest neighbors. Instead, here, we learn these features using similarities only, and then use a fast decoding algorithm on the learned features to output the final tree. Finally, HC has been studied in the model of parallel computation, where a variation of Boruvka's minimum spanning tree algorithm was used [6].

**Hyperbolic embeddings**  Hyperbolic geometry has been deeply studied in the network science community [34, 43], with applications to network routing for instance [23, 31]. Recently, there has been interest in using hyperbolic space to embed data that exhibits hierarchical structures. In particular, Sarkar [46] introduced a combinatorial construction that can embed trees with high fidelity, in just two dimensions (Fig. 1b). Sala et al. [45] extend this construction to higher dimensions and study the tradeoffs between precision and quality. Nickel et al. [41] propose a gradient-based method to embed taxonomies in the Poincaré model of hyperbolic space [41], which was further extended to the hyperboloid model [42]. More recently, hyperbolic embeddings have been applied to neural networks [27] and later extended to graph neural networks [13, 37] and knowledge graph embeddings [12] (see [11] for a survey). In contrast with all the above methods that assume a known input graph structure, HYPHC discovers and decodes a tree structure using pairwise similarities.

# B  Preliminaries

We first introduce our notational conventions in Appendix B.1 and define the notion of hyperbolicity in Appendix B.2. In Appendix B.3, we review some useful results of hyperbolic metrics that will be used throughout our proofs.

## B.1 Notations

**Binary trees** A *binary tree* is one that has all degrees either 1 (a leaf node) or 3 (internal node).[7] A binary tree with $n$ leaves has exactly $n-1$ internal nodes (of degree 3). A *rooted binary tree* is a binary tree such that one of its internal nodes (i.e. non-leaf node), the root, has degree exactly 2. Note that if we think of one leaf in a binary tree as the root, then removing it (and letting its unique neighbor be the new root) converts this into a *rooted binary tree* in the traditional sense (See Fig. 7c and Fig. 7d for examples of rooted and unrooted binary trees). By convention, we let $T$ denote any binary tree with leaves $1, \ldots, n$ and root 0 if its rooted.

**Tree and hyperbolic metrics** With $i, j, k$, we refer to nodes in $T$, and $z_i, z_j, z_k$ correspond to points in a hyperbolic embedding $Z = \{z_1, \ldots, z_n\} \subset \mathbb{B}_2^n$. With $o$ or $z_0$, we denote the origin of hyperbolic space and $d_{\mathbb{B}_2}$ is the hyperbolic metric; every tree $T$ defines a tree metric $d_T$. We overload $d(i,j) := d_T(i,j)$ and $d(z_i, z_j) := d_{\mathbb{B}_2}(z_i, z_j)$ when the types are clear. There is a close correspondence between hyperbolic and tree metrics, and we define the notion of quasi-isometries, which we will use throughout our proofs.

**Definition B.1** (Quasi-isometric embedding). *Let $T$ be a binary tree on $(n+1)$ leaves and $Z = \{z_0, \ldots, z_n\}$ an embedding set. The pair $(Z, T)$ is $(1 + \varepsilon, \kappa)$-quasi-isometric if:*

$$d(i,j) \leq d(z_i, z_j) \leq (1 + \varepsilon)d(i,j) + \kappa,$$

*for all $0 \leq i, j \leq n$.*

**Remark B.1.** *Note that Definition B.1 is equivalent to saying there is a quasi-isometric embedding from $\{z_0, \ldots, z_n\}$ to the root and leaves of $T$, i.e. the two metrics agree up to a linear transform. Compared to the usual definition of quasi-isometry, we consider only a one-sided version for convenience in the proofs.*

**Node depth** We overload $d_0(i) = d_T(0, i)$ to refer to the distance from node $i$ to the root in $T$, and $d_o(z_i) = d_{\mathbb{B}_2}(z_0, z_i)$ to be the distance to the origin in $\mathbb{B}_2$. Intuitively, $d_0(\cdot)$ and $d_o(\cdot)$ represent the "depth" of a node or point. Note that the depths are dependent on the choice of a base point as the root of a tree or origin of the space. We always use $o$ for $\mathbb{B}_2$ and the node indexed by 0 for trees.

**LCA** We let $i \vee j$ denote the LCA of two leaf nodes in $T$, and $z_i \vee z_j$ denote the hyperbolic LCA defined in Eq. (7). In particular, we say that $\{i, j | k\}_T$ holds if the LCA of $(i, j)$ has a larger depth than that of $(i, k)$ and $(j, k)$, i.e. $d_0(i \vee j) \geq \max\{d_0(i \vee k), d_0(j \vee k)\}$. Similarly, we say that $\{z_i, z_j | z_k\}_{\mathbb{B}_2}$ holds if $d_o(z_i \vee z_j) \geq \max\{d_o(z_i \vee z_k), d_o(z_j \vee z_k)\}$. We overload $\{i, j | k\}_T = \{i, j | k\}$ and $\{z_i, z_j | z_k\}_{\mathbb{B}_2} = \{z_i, z_j | z_k\}$ when the types are clear.

## B.2 Gromov's delta hyperbolicity

We define the Gromov [28] product which can be used to define $\delta$-hyperbolic spaces.

**Definition B.2** (Gromov product). *In any metric space $(X, d)$, the Gromov product of points $x, y \in X$ with respect to a third point $z \in X$ is:*

$$\langle x, y \rangle_z = \frac{1}{2} \left( d(x, z) + d(z, y) - d(x, y) \right).$$

When the base point $z$ is taken to be the origin of $\mathbb{B}_2$ or the root of a tree, we shorten this to $\langle x, y \rangle$ unambiguously.

A key characterization of hyperbolic spaces is the notion of $\delta$-hyperbolicity.

**Definition B.3** ($\delta$-hyperbolicity, four-point condition). *A metric space $(X, d)$ is $\delta$-hyperbolic if there exists $\delta \geq 0$ such that for all $w, x, y, z$ in $X$:*

$$\langle x, y \rangle_z \geq \min\{\langle x, w \rangle_z, \langle w, y \rangle_z\} - \delta.$$

**Example B.1.** *The hyperbolic space $\mathbb{B}_2$ is $\log 3$-hyperbolic.*

**Example B.2.** *Metric trees are 0-hyperbolic.*

**Example B.3.** *The Euclidean space $\mathbb{R}^n$ is not $\delta$-hyperbolic.*

(a) Euclidean triangle (not $\delta$-slim).

(b) 0-slim triangle in a tree.

(c) $\delta$-slim triangle.

Figure 4: Illustration of the notion of $\delta$-slim triangles.

Up to changing $\delta$ by a constant multiple, there are many equivalent variations of the notion of $\delta$-hyperbolicity. In particular, one intuitive interpretation of $\delta$-hyperbolic spaces is using the notion of $\delta$-slim triangles, which says that any triangle in a $\delta$-hyperbolic space has distance from any side to the other two less than $\delta$. This is not true in Euclidean space, for instance the midpoint of a large isosceles triangle might be far from the other two sides (Fig. 4).

## B.3 Tree-likeness of the hyperbolic space

An important result in the theory of hyperbolic metric spaces is their tree-likeness. We review two useful results of $\delta$-hyperbolic metrics. First, our notion of LCA depth is closely related to the Gromov product, which is exactly the tree depth for 0-hyperbolic metrics (Lemma B.1). Second, any finite set of points in $\mathbb{B}_2$ can be embedded in a binary tree (Proposition B.1):

**Lemma B.1** (LCA depth is close to the Gromov product). *For any $i, j \in T$,*

$$d_0(i \vee j) = \langle i, j \rangle.$$

*There exists $\delta > 0$ such that for any $z_i, z_j \in \mathbb{B}_2$,*

$$\langle z_i, z_j \rangle \leq d_o(z_i \vee z_j) \leq \langle z_i, z_j \rangle + \delta.$$

*Proof.* This is a direct application of Lemma 6.1 and 6.2 in [8] to tree metrics (which are 0-hyperbolic) and to the hyperbolic space $\mathbb{B}_2$ (which is $\delta$-hyperbolic).[8] □

**Proposition B.1** (Tree-likeness of hyperbolic space). *There is a constant $C_n$ such that for any set of points $\{z_0, z_1, \ldots, z_n\} \subset \mathbb{B}_2^{n+1}$, there is a binary tree $T$ on leaves $0, 1, \ldots, n$ such that:*

$$\forall 0 \leq i, j \leq n : d_T(i, j) \leq d_{\mathbb{B}_2}(z_i, z_j) \leq d_T(i, j) + C_n, \tag{12}$$

*with $C_n = \delta \cdot O(n)$.*

*Proof.* The statement of Proposition B.1 without the binary or leaf condition is a standard result (See Proposition 6.7 in [8]). That is, there exists a tree $T$ with $n$ nodes (not necessarily binary) satisfying Eq. (12). We modify $T$ to satisfy the leaf constraint (i.e. $\{z_0, z_1, \ldots, z_n\}$ are leaves' embeddings) and to satisfy the binary condition (i.e. $T$ is binary in the sense that every node has degree 1 or 3).

**Leaf condition** Let $k \in [n]$ be a node in $T$ and $m$ be the minimum edge length in $T$. If $k$ has degree greater than 1 (i.e. $k$ is not a leaf node), then we shrink every edge connected to it by some constant $c < \min\{\delta, \frac{m}{n}\}$, create a dummy node $p$ in place of $k$ and connect $k$ to it (Fig. 5a). Now all tree distances involving $k$ are the same, and all other distances going through $p$ are shrunk by at most $2c$. The resulting tree $T'$ has $n$ leaves and is such that:

$$\forall 0 \leq i, j \leq n : d_{T'}(i, j) \leq d_T(i, j) \leq d_{T'}(i, j) + 2c.$$

(a) Leaf condition.

(b) Binary tree condition.

Figure 5: Tree transformations used to satisfy the leaf and the binary conditions in Proposition B.1

**Binary condition** Next, we modify $T'$ to be binary. For every node of degree 2, simply delete it, which does not affect the tree metric restricted to $[n]$ (we could not have deleted a node in $[n]$ since they are all leaves now). For every node of degree 4 or more, we replace it by multiple copies connected by edges of length $c$ and decrease original edges by $c$ every time we create a copy of the original node (Fig. 5b), which causes the distances to shrink by at most $2c(n-1)$ (case of star trees). Therefore, distances in this new binary tree $T''$ satisfy:

$$\forall 0 \leq i, j \leq n : d_{T''}(i, j) \leq d_{T'}(i, j) \leq d_{T''}(i, j) + 2c(n-1).$$

With this final binary tree on $(n+1)$ leaves $T''$, we have:

$$\forall 0 \leq i, j \leq n : d_{T''}(i, j) \leq d_{\mathbb{B}}(z_i, z_j) \leq d_{T''}(i, j) + C_n + 2c + 2c(n-1).$$

Thus the statement holds for $C'_n = C_n + 2cn$, which is still $\delta \cdot O(n)$ since $c < \delta$. $\qquad \square$

**Remark B.2.** *Using the definition of quasi-isometries (Definition B.1), Proposition B.1 implies that for any embedding $Z \in \mathbb{B}_2^n$, there exists a binary tree $T$ on $n$ leaves such that there is a $(1, C_n)$-quasi-isometry from the leaves of $T$ to $Z$.*

## C Hyperbolic LCA construction

We detail the calculations used to compute the hyperbolic LCA and its distance to the origin.

**Lemma 4.1.** *Let $(x, y) \in (\mathbb{B}_2)^2$ and $x \vee y$ denote the point on the geodesic connecting $x$ and $y$ that minimizes the distance to the origin $o$. Let $\theta$ be the angle between $(x, y)$ and $\alpha$ be the angle between $(x, x \vee y)$. We have:*

$$\alpha = \tan^{-1}\left( \frac{1}{\sin(\theta)} \left( \frac{||x||_2(||y||_2^2 + 1)}{||y||_2(||x||_2^2 + 1)} - \cos(\theta) \right) \right),$$

$$\text{and } d_o(x \vee y) = 2 \tanh^{-1}(\sqrt{R^2 + 1} - R), \tag{8}$$

$$\text{where } R = \sqrt{\left( \frac{||x||_2^2 + 1}{2||x||_2 \cos(\alpha)} \right)^2 - 1}.$$

*Proof.* We use circle inversions to show this result. Circle inversions are Euclidean transformations on the plane that map circles to circles and preserve angles [9], and can represent hyperbolic reflections (isometric transformations) along hyperbolic geodesics. In particular, the circle inversion formula can be used to recover the center of the circle that is orthogonal to the boundary of the disk and that coincides with the geodesic between two given points. Consider the circle defined by the geodesic connecting $x$, and $y$ and let $R$ denote its radius, $p$ the orthogonal projection of $o$ onto this circle and $\Delta = \overline{oo'}$ denote the distance between the origin and the circle center (Fig. 6). By the circle inversion property and using the fact that the Poincaré disk has radius one, we have $1 = (\Delta - R)(\Delta + R)$, which yields:

$$R^2 = \Delta^2 - 1. \tag{13}$$

Additionally, if $\theta = \angle xoy$ and $\alpha = \angle pox$, we have by the Pythagorean theorem:

$$\begin{cases} R^2 = (\Delta - ||x||_2 \cos(\alpha))^2 + ||x||_2^2 \sin^2(\alpha) \\ R^2 = (\Delta - ||y||_2 \cos(\theta - \alpha))^2 + ||y||_2^2 \sin^2(\theta - \alpha). \end{cases} \tag{14}$$

Figure 6: Circle inversion used for hyperbolic LCA construction in Lemma 4.1.

This leads to the system of equations:

$$\begin{cases} 2\sqrt{R^2+1}\cos(\alpha) = \frac{||x||_2^2+1}{||x||_2} \\ 2\sqrt{R^2+1}\cos(\theta-\alpha) = \frac{||y||_2^2+1}{||y||_2} \end{cases} \tag{15}$$

which is solved for $R$ and $\alpha$ defined in Eq. (8). Now using Eq. (13), we have: $||p||_2 = \Delta - R = \sqrt{R^2+1} - R$. Finally, we get the result using the hyperbolic distance function and noting that the orthogonal projection of a point on a geodesic that does not contain that point is minimizing the distance between the point and the geodesic, that is $p = x \vee y$. $\qquad\square$

## D   Proof of Theorem 4.1

Our goal in this section is to show our main result, Theorem 4.1, which gives a $(1+\varepsilon)$-approximation ratio for Dasgupta's discrete objective. We first give an overview of the strategy used to show Theorem 4.1 and state the two results used in the proof (Lemma D.1 and Lemma D.2) in Appendix D.0. We then go over the details of Lemma D.1 in Appendix D.1 and Lemma D.2 in Appendix D.2.

### D.0   Proof outline

Our proof of Theorem 4.1 relies on two main results; the existence of a constrained embedding, the spread embedding set (Definition 4.1), such that the rooted binary tree decoded from any embedding in that set has a discrete cost close to the continuous cost (Lemma D.1), and reciprocally, any rooted binary tree has a corresponding embedding in that set (Lemma D.2).

First, we define the constrained spread embedding set and give more precise conditions on the spread constants, which we'll use throughout the proof.

**Definition D.1** (Spread Embeddings). *An embedding $Z \in \mathbb{B}_2^n$ is called* spread *if for every triplet $(i,j,k)$:*

$$\max\{d_o(z_i \vee z_j), d_o(z_i \vee z_k), d_o(z_j \vee z_k)\} - \min\{d_o(z_i \vee z_j), d_o(z_i \vee z_k), d_o(z_j \vee z_k)\} > 3C_n + 2\delta + 1, \tag{16}$$

*where $\delta$ is Gromov's delta hyperbolicity (Lemma B.1) and $C_n = \delta \cdot O(n)$ is defined in Proposition B.1.*

The spread constraints force LCAs to be distinguishable from each other, which intuitively encourages binary tree metrics. Now using this definition, we show that any spread embedding decoded using Algorithm 1 returns a tree that has a discrete cost close to the embeddings' continuous cost.

**Lemma D.1.** *Let $Z \in \mathcal{Z} \subset \mathbb{B}_2^n$ be a spread embedding. Then:*

$$|C_{\mathrm{Dasgupta}}(\mathsf{dec}(Z); w) - C_{\mathrm{HypHC}}(Z; w, \tau)| \leq 4e^{-1/\tau} \sum_{ijk} \max\{|w_{ij}|, |w_{ik}|, |w_{jk}|\}$$

We then show that any rooted binary tree has a corresponding spread embedding that decodes to it.

**Lemma D.2.** *For any unit-weight rooted binary tree $T$ on $n$ leaves, there exists a spread embedding $Z \in \mathcal{Z} \subset \mathbb{B}_2^n$ such that $\mathsf{dec}(Z) = T$.*

These two Lemmas show the tight equivalence between our continuous $C_{\text{HypHC}}$ cost on spread embeddings, and the discrete Dasgupta cost $C_{\text{Dasgupta}}$ on rooted binary trees. Finally, putting these together, we show our main result which is that the discrete tree returned by HYPHC has a $(1 + \varepsilon)$-approximation factor for Dasgupta's minimum (Theorem 4.1).

**Theorem 4.1.** *Consider a dataset with $n$ datapoints and pairwise similarities $(w_{ij})$ and let $T^*$ be the solution of Eq. (2). Let $\mathcal{Z}$ be the set of spread embeddings and $Z^* \in \mathcal{Z}$ be the solution of Eq. (4) for some $\tau > 0$. For any $\varepsilon > 0$, if $\tau \leq \mathcal{O}(1/\log(1/\varepsilon))$, then:*

$$\frac{C_{\text{Dasgupta}}(\mathsf{dec}(Z^*); w)}{C_{\text{Dasgupta}}(T^*; w)} \leq 1 + \varepsilon. \tag{11}$$

*Proof.* Let $\mathcal{Z} \subset \mathbb{B}_2^n$ be the set of spread leaves' embeddings embeddings (Definition D.1), $\tau > 0$ and:

$$T^* = \operatorname{argmin}_T C_{\text{Dasgupta}}(T; w)$$
$$Z^* = \operatorname{argmin}_{Z \in \mathcal{Z}} C_{\text{HypHC}}(Z; w, \tau).$$

WLOG, assume that all edges in $T^*$ have unit weight.[9] Since $T^*$ is a unit-weight rooted binary, we can apply Lemma D.2 to find $Z \in \mathcal{Z}$ such that $\mathsf{dec}(Z) = T^*$.

Next, let $\Delta := C_{\text{Dasgupta}}(\mathsf{dec}(Z^*); w) - C_{\text{Dasgupta}}(T^*; w)$. We have:

$$0 \leq \Delta \leq C_{\text{Dasgupta}}(\mathsf{dec}(Z^*); w) - C_{\text{HypHC}}(Z^*; w, \tau) + C_{\text{HypHC}}(Z; w, \tau) - C_{\text{Dasgupta}}(T^*; w)$$
$$= C_{\text{Dasgupta}}(\mathsf{dec}(Z^*); w) - C_{\text{HypHC}}(Z^*; w, \tau) + C_{\text{HypHC}}(Z; w, \tau) - C_{\text{Dasgupta}}(\mathsf{dec}(Z); w)$$
$$\leq 2 \sup_{Z \in \mathcal{Z}} |C_{\text{Dasgupta}}(\mathsf{dec}(Z); w) - C_{\text{HypHC}}(Z; w, \tau)|$$
$$\leq 8 \, e^{-1/\tau} \sum_{ijk} \max\{|w_{ij}|, |w_{ik}|, |w_{jk}|\}.$$

The first inequality follow from the fact that $C_{\text{HypHC}}(Z^*; w, \tau) \leq C_{\text{HypHC}}(Z; w, \tau)$ since $Z^*$ is the minimizer, and the last inequality uses Lemma D.1. Then:

$$\frac{C_{\text{Dasgupta}}(\mathsf{dec}(Z^*); w)}{C_{\text{Dasgupta}}(T^*; w)} \leq 1 + 8 \, e^{-1/\tau} \left( \frac{\sum_{ijk} \max\{w_{ij}, w_{ik}, w_{jk}\}}{C_{\text{Dasgupta}}(T^*; w)} \right). \tag{17}$$

Since $C_{\text{Dasgupta}}(T^*; w) \geq \sum_{ijk} \min\{w_{ij} + w_{ik}, w_{ij} + w_{jk}, w_{ik} + w_{jk}\} + 2 \sum_{ij} w_{ij}$, we finally have:

$$\frac{C_{\text{Dasgupta}}(\mathsf{dec}(Z^*); w)}{C_{\text{Dasgupta}}(T^*; w)} \leq 1 + \mathcal{O}(e^{-1/\tau}). \tag{18}$$

$\square$

**Remark D.1.** *At a high level, this bound suggests that a better approximation of the argmax function (lower $\tau$) gives a better approximation for Dasgupta's discrete objective.*

**Remark D.2.** *The optimization set $\mathcal{Z}$ defines a local constraint on every triplet $z_i, z_j, z_k$ (Definition 4.1), and can be enforced on triplets concurrently with sampling them for the main loss (Eq. (9)). The max and min operations can be relaxed using softmax, and the separation can be enforced with any auxiliary constraint (e.g. hinge loss).*

## D.1 Proof of Lemma D.1

The goal of this section is to show Lemma D.1. We first introduce a notion of LCA agreement (Appendix D.1.1). We show any spread embedding is in LCA agreement with some unrooted binary tree $T$, and gets decoded into the rooted version of $T$ (Lemma D.3 in Appendix D.1.2). This then allows us to show Lemma D.1 in Appendix D.1.3.

### D.1.1 LCA Agreement

We seek to understand when the continuous HYPHC cost of an embedding is close to the discrete cost of the tree decoded with Algorithm 1. The discrete and continuous costs both depend on the ordering of triplets, in terms of their LCAs' depths (i.e. distances to the origin or root). If this ordering is the same in the continuous embedding space and in the decoded discrete tree, then the costs should be close, up the the continuous approximation error of the softmax function. This motivates us to introduce the notion of LCA agreement.

**Definition D.2.** *Given a metric space $(X, d)$, and binary operation $\vee : X \times X \to X$, we define the set of* LCA triplets *on any points $x_1, \dots, x_n \in X$ with respect to the base point $x_0$ to be*

$$\{\{i, j|k\}, i, j, k \in [n] : d(x_0, x_i \vee x_j) > \min\{d(x_0, x_i \vee x_k), d(x_0, x_j \vee x_k)\}.$$

**Definition D.3** (LCA agreement). *We say two sets of points $x_1, \dots, x_n \in X^n$ and $y_1, \dots, y_n \in Y^n$ (possibly in different metric spaces) are in* LCA agreement *or* LCA equivalent *with respect to $x_0$ and $y_0$, if their set of LCA triplets are identical.*

Note that LCA agreement is a relative notion and depends on the base point. In our proofs, we always refer to LCA agreement as being with respect to the origin $z_0 = o \in \mathbb{B}_2$ for embeddings, and with respect to the node labelled 0 for trees.

Intuitively, LCA agreement implies that the LCAs of an embedding are consistent with a ground truth tree. Since our proposed decoding algorithm only relies on LCA distances, we can show that if there exist a unrooted binary tree $T$ such that $(Z, T)$ are in LCA agreement, then $\text{dec}(Z)$ recovers the rooted version of $T$.

**Lemma D.3.** *Let $Z$ be an embedding and $T$ be a binary tree on $(n + 1)$ leaves (not rooted) such that $(Z \cup \{z_0 = o\}, T)$ are in LCA agreement. Let $T'$ be the rooted binary tree on $n$ leaves that is obtained by removing the leaf $0$ from $T$. Then $\text{dec}(Z) = T'$.*

*Proof.* To be a little more formal, define $\text{dec}(\cdot)$ to depend on three things: a set of points ($Z$ or $T$), a base point ($z_0 = o$ or 0), and a LCA construction ($\vee_{\mathbb{B}_2}$ or $\vee_T$).

Let $T'$ be the rooted binary tree that is obtained by removing the leaf 0 from $T$, and relabelling its neighbor $r$ in $T$ as 0, the root of $T'$ (see Fig. 7c and Fig. 7d). Because $Z$ and $T$ are in LCA agreement, the tree constructed from $\text{dec}(Z; o, \vee_{\mathbb{B}_2})$ is exactly the same as the tree constructed from $\text{dec}(T; 0, \vee_T)$. Thus we only have to show that $T' = \text{dec}(T; 0, \vee_T)$.

We use an induction argument to show that $T' = \text{dec}(T; 0, \vee_T)$. Let $r$ be the unique neighbor of 0 in $T$, and let $T_0$ and $T_1$ be the two trees on the other edges of $r$ aside from the one pointing to 0 (Fig. 7e). Note that any distances $d_T(0, i \vee j)$ for any $i, j$ both contained in $T_0$ or $T_1$ are strictly smaller than distances $d_T(0, i \vee j)$ for any $i \in T_0, j \in T_1$. Therefore the decoding algorithm merges all pairs $(i, j) \in T_0$ and $(i, j) \in T_1$ first. Inductively, this decoding algorithm will exactly return $T_0$ and $T_1$.

Finally, the decoding algorithm will merge any two $i \in T_0, j \in T_1$. This creates a new parent and connects them to the roots of $T_0, T_1$ (Algorithm 1, line 7). The structure of this tree is therefore the same as the tree rooted at $r$, in other wise $T'$, as desired. $\square$

### D.1.2 Any spread embedding is in LCA agreement with some binary tree

Lemma D.3 shows that under the LCA agreement condition, the decoding algorithm will produce a discrete tree that preserves the ordering of triplets (in terms of LCAs' depths), thus the continuous and discrete costs are close. Furthermore, we know by Proposition B.1 that any embedding has a corresponding quasi-isometric binary tree. Using this, we seek a condition under which a quasi-isometric pair $(Z, T)$ is in LCA agreement.

First, we show that under a quasi-isometric embedding (Definition B.1), the hyperbolic LCA depth $d_o(z_i \vee z_j)$ is close to the tree LCA depth $d_0(i \vee j)$ (Lemma D.4). Next, we derive a condition under which a quasi-isometric pair $(Z, T)$ is in LCA agreement (Lemma D.5). Finally, we show that any spread embedding satisfies this condition, and more specifically, we show that any spread embedding is in LCA agreement with some binary tree, such that the "deepest" LCA is always distinguishable from the embeddings (Lemma D.6).

Figure 7: (a): Leaves embeddings and the origin in $\mathbb{B}_2$. (b): Binary tree that is a quasi-isometry of the leaves embeddings and the origin (Proposition B.1) and corresponding discrete binary tree (not rooted) in (c). (d): Rooted binary tree obtained by decoding hyperbolic leaves embeddings. Observe that the quasi-isometric tree in (c) and the decoded tree in (d) are LCA equivalent with respect to 0. (e): Figure used in induction proof in Lemma D.3.

**Lemma D.4** (Hyperbolic LCA depth is close to Tree LCA depth). *Let $Z \subset \mathbb{B}_2^{n+1}$ be an embedding with $z_0 = o$ and $T$ a binary tree (not rooted) on $(n+1)$ leaves such that $(Z, T)$ is $(1 + \varepsilon, \kappa)$-quasi-isometric. If $M$ is the diameter of $T$, then for any leaves $1 \leq i, j, k, l \leq n$:*

$$|(d_o(z_i \vee z_j) - d_o(z_k \vee z_l)) - (d_0(i \vee j) - d_0(k \vee l))| \leq \frac{3\varepsilon}{2}M + \frac{3}{2}\kappa + \delta. \qquad (19)$$

*Proof.* We bound the hyperbolic LCA depth by the corresponding tree depths:

$$
\begin{aligned}
d_o(z_i \vee z_j) &\geq \langle z_i, z_j \rangle \\
&= \frac{1}{2}\left(d_o(z_i) + d_o(z_j) - d(z_i, z_j)\right) \\
&\geq \frac{1}{2}\left(d_0(i) + d_0(j) - (1 + \varepsilon)d(i,j) - \kappa\right) \\
&= d_0(i \vee j) - \frac{\varepsilon}{2}d(i,j) - \frac{1}{2}\kappa.
\end{aligned}
$$

The first line is Lemma B.1, the second line is Definition B.2, and the third applies Definition B.1.

In the other direction, again using Lemma B.1, Definition B.2, and Definition B.1:

$$
\begin{aligned}
d_o(z_i \vee z_j) &\leq \langle z_i, z_j \rangle + \delta \\
&= \frac{1}{2}\left(d_o(z_i) + d_o(z_j) - d(z_i, z_j)\right) + \delta \\
&\leq \frac{(1+\epsilon)\left(d_0(i) + d_0(j)\right) + 2\kappa}{2} - \frac{d(i,j)}{2} + \delta \\
&\leq \frac{1}{2}\left(d_0(i) + d_0(j) - d(i,j)\right) + \frac{\epsilon}{2}\left(d_0(i) + d_0(j)\right) + \kappa + \delta \\
&= d_0(i \vee j) + \frac{\varepsilon}{2}d_0(i) + \frac{\varepsilon}{2}d_0(j) + \kappa + \delta.
\end{aligned}
$$

Finally, Eq. (19) follows by adding these inequalities with $d(i,j) \leq M$ for any $0 \leq i, j \leq n$. $\qquad \square$

Lemma D.4 allows us to provide a concrete condition for when $(Z, T)$ are in LCA agreement, which will be our main tool for showing the consistency of our relaxation.

**Lemma D.5.** *Let $Z \subset \mathbb{B}_2^{n+1}$ be an embedding with $z_0 = o$ and $T$ a binary tree (not rooted) on $(n+1)$ leaves, such that $(Z, T)$ is $(1 + \varepsilon, \kappa)$-quasi-isometric. Define $M$ to be the diameter of $T$ and $m$ to be the length of the smallest edge not including a leaf of $T$. If $m > \frac{3\varepsilon}{2}M + \frac{3}{2}\kappa + \delta$, then $(Z, T)$ are in LCA agreement.*

*Proof.* Consider an arbitrary triple $(i, j, k)$, and suppose $\{ij|k\}_T$ holds. It suffices to show that $\{z_i, z_j|z_k\}_{\mathbb{B}_2}$ holds to show that $(Z, T)$ are in LCA agreement (Definition D.3). Applying

Lemma D.4,

$$d_o(z_i \vee z_j) - d_o(z_i \vee z_k) \geq d_0(i \vee j) - d_0(i \vee k) - \left( \frac{3\varepsilon}{2} M + \frac{3}{2}\kappa + \delta \right)$$

$$\geq m - \frac{3\varepsilon}{2} M - \frac{3}{2}\kappa - \delta > 0.$$

In the second line, we used the fact that since $i \vee j$ and $i \vee k$ are both internal nodes in $T$, their distance is at least $m$ by assumption (also, $i \vee j$ is deeper than $i \vee k$ by assumption, so the sign of the difference is positive). Similarly $d_o(z_i \vee z_j) - d_o(z_j \vee z_k) > 0$, so $z_i \vee z_j$ is the deepest out of the 3 LCAs, as desired. □

Lemma D.5 gives a technical condition for when an embedding $Z$ is in LCA agreement with a tree $T$. Next, we claim that the constrained set of spread embeddings $\mathcal{Z} \subseteq \mathbb{B}_2^n$ is such that for every embedding $Z \in \mathcal{Z}$, there is always a corresponding tree $T$ satisfying Lemma D.5. Intuitively, Lemma D.5 says that $m$ should be large, i.e. the internal edges of the tree should be spread far apart. Using our notion of hyperbolic LCA, we can codify this by enforcing that different LCAs should be far from each other. By leveraging global properties of hyperbolic space, this is in fact sufficient.

**Lemma D.6.** *Suppose that embedding $Z \in \mathcal{Z} \subset \mathbb{B}_2^n$ is spread. Then there exist a binary tree (not rooted) on $(n+1)$ leaves $T$ such that $(Z \cup \{z_0 = o\}, T)$ are in LCA agreement. In particular, for any $1 \leq i, j, k \leq n$ such that $\{i, j|k\}_T$ holds, then:*

$$d_o(z_i \vee z_j) > \max\{d_o(z_i \vee z_k), d_o(z_j \vee z_k)\} + 1.$$

*Proof.* Append the origin $z_0$ to $Z$ (so we have a collection of $n+1$ points). By Proposition B.1, there is a binary tree $T$ (not rooted) on $(n+1)$ leaves that is $(1, C_n)$-quasi-isometric for $Z \cup \{z_0\}$, where $C_n = \delta \cdot \mathcal{O}(n)$. Consider an internal edge $e \in T$, i.e. an edge connecting non-leaf nodes. Since $T$ is binary, $e$ has endpoints $i \vee j$, $i \vee j \vee k$ for some triplet $\{ij|k\}_T$ with $1 \leq i, j, k \leq n$.[10]

Let $a, b, c$ be the ordering of $i, j, k$ such that $d_o(z_a \vee z_b) > d_o(z_b \vee z_c) > d_o(z_a \vee z_c)$. Consider:

$$d_0(a \vee b) - d_0(a \vee c) \geq d_o(z_a \vee z_b) - d_o(z_a \vee z_c) - \left( \frac{3}{2} C_n + \delta \right)$$

$$> \frac{3}{2} C_n + \delta + 1,$$

where the first line applies Lemma D.4 with $\varepsilon = 0, \kappa = C_n$, and the second uses the definition of spread (Definition D.1), which holds for any $1 \leq a, b, c \leq n$. In particular, $d_0(a \vee b) - d_0(a \vee c) > 0$, so clearly $\{a, b|c\}_T$ holds since $T$ is binary, and we must have:

$$d_0(a \vee b) - d_0(a \vee c) = d_0(i \vee j) - d_0(i \vee j \vee k) = d(i \vee j, i \vee j \vee k),$$

which is the length of the edge $e$ we are considering. Since this holds generically for any edge $e$ among internal nodes of $T$, this also holds for the minimum edge length $m$:

$$m > \frac{3}{2} C_n + \delta + 1. \tag{20}$$

In particular, the conditions of Lemma D.5 apply; that is, $m$ satisfies $m > \frac{3\varepsilon}{2} M + \frac{3}{2}\kappa + \delta$ for $\varepsilon = 0, \kappa = C_n$. Applying Lemma D.5, we get that $(Z \cup \{z_0\}, T)$ are in LCA agreement.

We now turn to the second part of Lemma D.6 which bounds the difference in LCA depth. Let $1 \leq i, j, k \leq n$ such that $\{i, j|k\}_T$ holds. Since $(Z, T)$ are in LCA agreement, we can assume WLOG that $d_o(z_i \vee z_j) > d_o(z_i \vee z_k) > d_o(z_j \vee z_k)$. Applying Lemma D.4 again, we have:

$$d_o(z_i \vee z_j) - d_o(z_i \vee z_k) \geq d_0(i \vee j) - d_0(i \vee k) - (\frac{3}{2} C_n + \delta)$$

$$\geq m - \frac{3}{2} C_n - \delta$$

$$> 1,$$

where we used Eq. (20) in the last inequality. □

### D.1.3 The tree decoded from any spread embedding has a cost close to the HYPHC cost

We now have all the tools to show that any spread embedding decodes to a tree such that the discrete and continuous costs are close (Lemma D.1).

**Lemma D.1.** *Let $Z \in \mathcal{Z} \subset \mathbb{B}_2^n$ be a spread embedding. Then:*

$$|C_{\text{Dasgupta}}(\text{dec}(Z); w) - C_{\text{HYPHC}}(Z; w, \tau)| \leq 4e^{-1/\tau} \sum_{ijk} \max\{|w_{ij}|, |w_{ik}|, |w_{jk}|\}$$

*Proof.* Let $Z \in \mathcal{Z} \subset \mathbb{B}_2^n$ be a spread embedding. Using Lemma D.6, we know that there exists a binary tree (not rooted) on $(n + 1)$ leaves $T$, such that $(Z \cup \{z_0 = o\}, T)$ are in LCA agreement. Let $T'$ be the rooted binary tree that is obtained by removing the leaf 0 from $T$, and relabelling its neighbor $r$ in $T$ as 0, the root of $T'$. Using Lemma D.3, we know that $T' = \text{dec}(Z)$ and:

$$|C_{\text{Dasgupta}}(\text{dec}(Z); w) - C_{\text{HYPHC}}(Z; w, \tau)| \leq \sum_{ijk} |w_{ijk}(T'; w) - w_{\text{HYPHC}, ijk}(Z; w, \tau)|.$$

Let $\delta_{ijk} \coloneqq |w_{ijk}(T'; w) - w_{\text{HYPHC}, ijk}(Z; w, \tau)|$. WLOG, assume that $\{i, j|k\}_{T'}$ holds for a triplet $(i, j, k) \in T'$. $T$ and $T'$ are equivalent in the LCA agreement sense with respect to 0, since LCA agreement in defined over leaves in $[n]$. That is, the LCA of any pair $i, j \in [n]$ with respect to 0 in $T$ is the same as the LCA with respect to 0 in $T'$. Therefore using the definition of LCA agreement, we have:

$$d_1 \coloneqq d_o(z_i \vee z_j) \geq \max\{d_o(z_i \vee z_k), d_o(z_j \vee z_k)\} \coloneqq d_2.$$

Denote $w^*_{ijk} \coloneqq \max\{|w_{ij}|, |w_{ik}|, |w_{jk}|\}$ and $\Sigma_{ijk} \coloneqq e^{d_o(z_i \vee z_j)/\tau} + e^{d_o(z_i \vee z_k)/\tau} + e^{d_o(z_j \vee z_k)/\tau}$. Then:

$$\begin{aligned}
\delta_{ijk} &= \left| w_{ij}\left(1 - \frac{e^{d_o(z_i \vee z_j)/\tau}}{\Sigma_{ijk}}\right) + w_{ik}\left(\frac{e^{d_o(z_i \vee z_k)/\tau}}{\Sigma_{ijk}}\right) + w_{jk}\left(\frac{e^{d_o(z_j \vee z_k)/\tau}}{\Sigma_{ijk}}\right) \right| \\
&\leq 2w^*_{ijk}\left(\frac{e^{d_o(z_i \vee z_k)/\tau} + e^{d_o(z_j \vee z_k)/\tau}}{\Sigma_{ijk}}\right) \qquad\qquad (21) \\
&\leq 4w^*_{ijk}e^{(d_2 - d_1)/\tau} \\
&\leq 4w^*_{ijk}e^{-1/\tau}.
\end{aligned}$$

In the last line, we applied the second part of Lemma D.6. $\qquad\square$

### D.2 Proof of Lemma D.2

We have shown in Lemma D.6 that any spread embedding gets decoded into a tree such that the continuous and discrete costs are close. We now show the other direction, that any rooted binary tree has a corresponding spread embedding which decodes back to it (Lemma D.2), and therefore the discrete and continuous costs are close.

We first recall a result by Sarkar for low-distortion hyperbolic embeddings of trees [46].

**Proposition D.1.** *Any unit-weight tree $T$ can be embedded into $Z$ with scale $\zeta = O(1/\varepsilon)$ and worst-case distortion at most $1 + \varepsilon$, i.e.*

$$\zeta d_T(i, j) \leq d(z_i, z_j) \leq \zeta(1 + \varepsilon)d_T(i, j).$$

In our terminology, this says that if every edge of $T$ is weighed with a scalar $\zeta = \mathcal{O}(1/\varepsilon)$, then there is an embedding $Z$ such that $(Z, T)$ is $(1 + \varepsilon, 0)$-quasi-isometric, that is:

$$d_T(i, j) \leq d(z_i, z_j) \leq (1 + \varepsilon)d_T(i, j).$$

We now rely on Sarkar's result to find a spread embedding for a given rooted binary tree.

**Lemma D.7.** *Let $T$ be any unit-weight binary tree (not rooted) on $(n + 1)$ leaves. Then there is a spread embedding $Z \in \mathcal{Z} \subset \mathbb{B}_2^{n+1}$ with $z_0 = 0$, such that $(Z, T)$ are in LCA agreement.*

(a) First split (red).     (b) Second split (blue).     (c) Third split (green).

Figure 8: An illustration of the greedy decoding algorithm.

*Proof.* Let $\varepsilon > 0$. Put a weight $\zeta$ ($\zeta$ to be decided later) on every edge of $T$. Using Proposition D.1, embed $T$ to an embedding $Z \subset \mathbb{B}_2^{n+1}$, such that $(Z, T)$ is $(1 + \varepsilon, 0)$-quasi-isometric, and WLOG reflect the embeddings (isometric transformation) so that $z_0$ is at the hyperbolic origin.

Consider any triplet $1 \le i, j, k \le n$, and WLOG let $\{ij|k\}_T$. Applying Lemma D.4 with $\kappa = 0$,

$$d_o(z_i \vee z_j) - d_o(z_i \vee z_k) \ge d_0(i \vee j) - d_0(i \vee k) - \frac{3\varepsilon}{2}\zeta n - \delta$$

$$\ge \zeta\left(1 - \frac{3\varepsilon}{2}n\right) - \delta$$

In the second line we used the fact that the diameter of $T$ is at most $\zeta n$ (since there are $n + 1$ nodes and all edges are equally weighted), and in the third that $i \vee j \ne i \vee k$ and the minimum tree distance between any distinct nodes is $\zeta$.

Finally, we could choose $\varepsilon \le \frac{1}{3n}$ and $\zeta > 6C_n + 6\delta + 2$. Note that choosing such $\zeta = \Theta(n)$ works since Proposition D.1 says $\zeta = \Theta(1/\varepsilon) = \Theta(n)$ is possible for the embedding, and Proposition B.1 says it is sufficient since $C_n = O(n)$. Then:

$$\max\{d_o(z_i \vee z_j), d_o(z_i \vee z_k), d_o(z_j \vee z_k)\} - \min\{d_o(z_i \vee z_j), d_o(z_i \vee z_k), d_o(z_j \vee z_k)\}$$
$$\ge d_o(z_i \vee z_j) - d_o(z_i \vee z_k)$$
$$\ge \zeta\left(1 - \frac{3\varepsilon}{2}n\right) - \delta$$
$$> 3C_n + 2\delta + 1.$$

Since $i, j, k$ were arbitrary, this shows that $Z$ is spread (as defined in Definition D.1). Finally, note that the min edge length of $T$ is $m = \zeta$, the maximum path length is at most $M = n\zeta$, and $\zeta > \frac{3}{2}\varepsilon(n\zeta) + \delta$ by choice of $\varepsilon$ and $\zeta$. Since $(Z, T)$ were a $(1 + \epsilon, 0)$-quasi-isometry, Lemma D.5 then implies that $(Z, T)$ are in LCA agreement. □

Finally, we show Lemma D.2 using the previous Lemma and Lemma D.3.

**Lemma D.2.** *For any unit-weight rooted binary tree $T$ on $n$ leaves, there exists a spread embedding $Z \in \mathcal{Z} \subset \mathbb{B}_2^n$ such that $\mathrm{dec}(Z) = T$.*

*Proof.* Attach a leaf 0 to the root of $T$, apply Lemma D.7, then apply Lemma D.3. □

# E Experimental details

## E.1 More experimental details

**Datasets** In our experiments, we compute similarities using the cosine similarity measure on the datapoints' features. For all datasets in the UCI machine learning repository, we use the available features and normalize them so that each attribute has mean zero and standard deviation one. For CIFAR-100, where the raw data comes in the form of images, we used a pretrained BiT [33] convolutional neural network to compute 2048-dimensional image features (one to last layer).

| | | Zoo | Iris | Glass | Segmentation | Spambase |
|---|---|---|---|---|---|---|
| | # Points | 101 | 150 | 214 | 2310 | 4601 |
| | # Clusters | 7 | 3 | 6 | 7 | 2 |
| Discrete | SL | 97.7 | 76.7 | 50.3 | 51.1 | 61.2 |
| | AL | 90.1 | 73.7 | 46.3 | 58.2 | 73.4 |
| | CL | 96.6 | 76.1 | 46.9 | 55.1 | 69.1 |
| | WL | 90.0 | 74.9 | 48.3 | 61.3 | 68.3 |
| | BKM | 86.5 | 66.1 | 43.5 | 57.2 | 74.9 |
| Continuous | UFit | 97.2 | 76.8 | **51.0** | **61.2** | 61.6 |
| | gHHC | - | - | 46.3 | - | 61.4 |
| | HYPHC | **98.7** | **77.3** | 49.2 | 55.9 | **79.2** |

Table 2: Clustering quality results measured in dendrogram purity (DP). Best score in bold, second best underlined. gHHC scores are directly taken from [38].

**Baselines**    To compute numbers for agglomerative clustering methods, we used the corresponding implementation in the scipy Python library.[11] We implemented our own version of BKM following the description in [40] since no open-source version was available. For UFit, we used the open-source implementation and reused the same hyper-parameters as in the original paper [19].[12]

**Evaluation metrics**    Dasgupta's cost is a well-studied objective with known guarantees when there is an underlying ground-truth hierarchy [21] (such results have not been established for metrics like dendrogram purity (DP) [30]). We therefore measure the clustering quality in terms of the discrete Dasgupta Cost.[13] For randomized algorithms and our method (which produce a different solution for every run), we report the best cost over five random seeds. This is standard since all methods can be viewed as *search* algorithms for the latent minimizer of the Dasgupta cost, analogous to how standard combinatorial search algorithms for NP-hard problems rely on random restarts and global perturbations when they reach local optima.

**Cost Bounds**    We also report upper and lower bounds on the discrete Dasgupta cost, computed as:

$$\mathrm{UB}(w) \coloneqq \sum_{ijk} \max(w_{ij} + w_{ik}, w_{ij} + w_{jk}, w_{ik} + w_{jk}) + 2\sum_{ij} w_{ij} \geq C_{\mathrm{Dasgupta}}(T; w)$$

$$\mathrm{LB}(w) \coloneqq \sum_{ijk} \min(w_{ij} + w_{ik}, w_{ij} + w_{jk}, w_{ik} + w_{jk}) + 2\sum_{ij} w_{ij} \leq C_{\mathrm{Dasgupta}}(T; w),$$

(22)

for all rooted binary tree $T$. For datasets with more than a thousand of nodes, we sample triplets uniformly at random for both the lower and upper bounds, and report the average over 10 random seeds.

Note that datasets where the relative gap between the upper and lower bound is larger indicate datasets where similarities induce a more hierarchical structure. As noted in [50], an instance for which the lower bound can be achieved by a tree, is a "perfect" HC instance (termed "perfect HC-structure" in [50]), in the sense that for every three points, the tree decomposes them in the most preferred way, i.e., by cutting the highest similarity weight last, towards the bottom of the tree. As the optimum tree gets higher costs, approaching the upper bound above, the instance loses its hierarchical structure; for example, if the given graph is a unit weight clique with no hierarchy to be found, both upper and lower bounds coincide and actually this implies that any tree gets the same cost, as was shown in Dasgupta [25].

In Table 1, we note that on dataset where there is a large relative gap between the upper and lower bounds (e.g. Iris or Segmentation), the relative improvement of HYPHC compared to the best baseline

is more important compared to datasets with a smaller gap (e.g. CIFAR-100). This suggest that when there is a good underlying HC in the data, HYPHC is able to get closer to it than heuristic algorithms.

## E.2 Greedy decoding

To provide more intuition about the greedy decoding, we illustrate different steps of greedy decoding on a small example in Fig. 8. The first split is computed using the two largest angles splits (red lines in Fig. 8a). Then, the algorithm recurses on the two created subsets and uses the largest angle in each subset to split the data (blue and green lines in Fig. 8b and Fig. 8c).

## E.3 Comparison with gHHC

**Models' comparison** For completeness, we discuss in more details the comparison between HY-PHC and the gHHC model. Monath et al. [39] propose a differentiable HC objective, which yields improvements in scalability and downstream task performance. This model learns representations for a fixed number of intermediate nodes using hyperbolic embeddings, and optimizes such embeddings for the HC task. Once learned, the embeddings can be decoded into a discrete tree using heuristics and post-processing rules. gHHC differs from the traditional similarity-based HC setting, assuming additional information about the optimal clustering, in the form of hyperbolic leaves' embeddings. These are computed using normalized Euclidean features, which is mismatched to the geometry of the data. In contrast, HYPHC does not rely on input leaves' embeddings and directly optimizes the entire tree structure, via the hyperbolic LCA construction. The HYPHC decoding does not require post-processing and directly produces a dendrogram which matches the underlying geometry of the embeddings.

**Experiments** While we compare to similarity-based HC methods in our main experiments, we also include a comparison to gHHC (which uses features) for completeness in Table 2. gHHC evaluates the clustering quality using the dendrogram purity (DP) measure [30]. Given ground truth flat clusters, DP measures how well the clusters are preserved in the hierarchy. Note that this metric will only be a good indicator for clustering quality when the ground truth flat clusters correlate with the hierarchy. A proxy to measure such a correlation is analyzing if methods that do well on DC also do well on DP. On Spambase, HYPHC and BKM have the best DC and DP scores, while SL does poorly for both metrics. This suggests that ground truth flat clusters do correlate well with the optimal hierarchy on this dataset, and we note that HYPHC significantly outperforms gHHC for the DP metric. On Glass, the correlation is not so obvious; for instance SL performs poorly on DC but does very well on DP. We conjecture that the ground truth flat clusters in glass are not directly correlated with the optimal clustering on this dataset.

## Footnotes

[7]We use undirected trees in our proofs.

[8]Note that the notion of $\delta$-hyperbolicity in [8] uses the incenter condition, which is equivalent to the four point condition (Definition B.3), up to changing $\delta$ by a constant multiple. In standard hyperbolic space, Lemma B.1 is in fact satisfied for $\delta = \log 3$.

[9]Any weighted version of $T^*$ would achieve the same Dasgupta cost, so we consider the unit weight case for simplicity.

[10]Note that triplets here are defined on leaves $[n]$, excluding 0. The reason is that, since the LCA is computed with respect to 0 which is a leaf in $T$, $0 \vee i = 0 \ \forall i$, and therefore there is no internal edge whose endpoint is an LCA on 0.

[11]https://docs.scipy.org/doc/scipy/reference/cluster.hierarchy.html

[12]https://github.com/PerretB/ultrametric-fitting

[13]For completeness, we also report DP scores in Table 2, observing that this metric does not correlate well with DC on some datasets (see Appendix E.3 for a discussion of the results).