[Reviews · NeurIPS 2020]

Review 1

Summary and Contributions: I have read author response and updated my score. The authors propose a method for embedding datapoints into hyperbolic space by optimizing the proposed continuous relaxation of Dasgupta’s cost function for hierarchical clustering. Key contributions include coming up with relaxing the discrete cost into a continuous version using properties of hyperbolic spaces (and hence being amenable to gradient descent based optimization), and decoding a clustering from learnt representations in hyperbolic space. They also provide a approximation guarantees for the proposed method under perfect optimization assumptions, and the proposed method is competitive with other standard clustering algorithms. But I feel that there are a few missing baselines/experiments which might be critical to decide the fate of the work.

Strengths: - Provide approximation guarantees for the proposed method under some assumptions wrt optimization - Empirical results support the claims made in the paper but there are some missing baselines which I talk about in Weakness section - This work extends on gradient descent based methods for clustering which in my opinion are useful for scaling to large datasets, and the ideas presented in the paper will trigger some good conversation!

Weaknesses: - Overall, I buy the argument using gradient descent based methods for clustering allows scaling to large dataset. But the largest dataset that the authors use is about 50K points, which is maybe still not “large enough”. - For clustering experiments, authors evaluate all methods wrt Dasgupta’s cost. But all of those datasets also have ground-truth clustering, and it might be useful to evaluate proposed approach using some standard clustering evaluation metrics using ground-truth clusters. Currently, it is difficult to understand how much improvement in Dasgupta’s cost function is significant. - Also authors do not compare with top-down divisive clustering methods even though they themself use a similar method for decoding trees, and top-down divisive clustering methods usually have better time complexity than agglomerative methods that the authors compare with, and those methods also provide a good approximation of Dasgupta’s cost function. - Missing baseline for sec 6.4 (end-to-end training). It would be nice to see a comparison with classifiers that do not use any clustering step to see how some sort of clustering would help, even when used in a two-step cluster-then-classify fashion.

Correctness: There are some missing baselines in the experiments which I have talked about in Weaknesses. Also, I am not confident about the end-to-end training results (Sec 6.4) but that might be because of my poor understanding or missing details.

Clarity: The paper is reasonably well written. However, the authors point to the supplementary material several times, and it might be better to include some more examples, tables from supplementary material into the main paper. For ex, it might be better to provide expression to find hyperbolic LCA in the main paper (the derivation could be deferred to supplementary material though)

Relation to Prior Work: Yes, contributions on top of previous work are clearly stated.

Reproducibility: Yes

Additional Feedback: Q1.In the end-to-end training section, do the authors learn embeddings by clustering all points together? As in are train, test, and dev points all clustered together or are each of them clustered separately? If all the points are clustered separately then it might not be a reasonable thing in practice because in practice, we do not have access to test data while training, and nor should any test data be used for doing any sort of training. If authors perform some clustering on test points as well, then it might not be reasonable to assume access to *all* test data at test time. Evaluation on test data should preferably be possible even when test data arrives in an online fashion. If authors only cluster train datapoints, then how do they get representations for test data points? Also, how well does a simple classifier (without any clustering) do? Q2. The greedy decoding does something like 2-means (or k-means) Hierarchical clustering. I am curious how well does k-means HC does on these datasets? Q3: How well does decoding using the proposed approach work when the leaf embeddings are obtained by unit-norming input embeddings so that they lie of the hyperbolic diameter? This would be something like using leaf embeddings from gHHC paper but using the proposed tree decoding method. I don’t expect it to do great but it would be nice to understand how much improvement do we get by learning the leaf embeddings.


Review 2

Summary and Contributions: The paper introduces the use of hyperbolic embeddings for similarity-based hierarchical clustering. The paper shows that the embedding distorts the optimal solution by at most a (1+epsilon) factor. Moreover, the paper gives an explicit recovery algorithm that translates an optimal solution in the embedded space to a hierarchical clustering tree to the input space that is within (1+epsilon) factor of the true optimum.

Strengths: Although hyperbolic embeddings have been used for better embeddings on trees than on Euclidean spaces, this paper provably shows that hyperbolic embeddings (and the corresponding recovery algorithm) distorts the optimal solution by a (1+epsilon) factor. The explicit recovery algorithm given by this paper is also important. Also, hierarchical clustering is an important problem that is relevant to the scope of NeurIPS.

Weaknesses: The analysis only demonstrates that the optimal solution in the embedded space translates to a (1+epsilon) approximation in the original space, under the recovery algorithm; it is unclear how approximations to the optimal solution in the embedded space perform. Similarly, there are no approximation algorithms to the optimal solution in the embedded space that are given, though these should be more feasible since the embedded space is now continuous.

Correctness: Although the properties of hyperbolic embeddings were amply outlined, I didn't understand how each point is mapped under such an embedding. Otherwise the high-level ideas seem reasonable to me.

Clarity: Since hyperbolic embedding is still a relatively new tool, more details or examples for the actual mapping of each point in such embeddings would have been appreciated. Otherwise, I thought the paper is well-written.

Relation to Prior Work: There is thorough discussion on related work and their differences from this work.

Reproducibility: Yes

Additional Feedback: A correct understanding of how each point (leaf node) in the original space is mapped to some corresponding point in the hyperbolic embedding would help the confidence in my evaluation of this paper. =======Post-rebuttal update======= Although the embedding provides a (1+eps) distortion, the feedback states that there are no known improvements for approximating the continuous optimum, in which case there does not seem to be a provable advantage to using the embedding. Nevertheless, the paper gives a nice novel proof-of-concept. In the experimental sections, the authors compare the greedy and sampling heuristics in the continuous space to other baselines in the original space. However, the improvement of the continuous embedding approach over the other baselines is not convincing to me, given the relativity of the objective values, hyperparameter tuning, and the total number of runs.


Review 3

Summary and Contributions: The paper presents an approach for hierarchical clustering in hyperbolic space. The starting point is the work presented in [12], a discrete cost function for the hierarchical clustering over binary trees. The main contribution of the paper is to propose a relaxation of the cost through hyperbolic space leading to a differentiable cost which allows to find an hyperbolic embedding of the data to cluster. A decoding algorithm is then used to retrieve the corresponding discrete binary tree from the embeddings. Theoretical results mainly concern the correctness of the relaxation wrt the optimal discrete cost. Experiments are conducted on real datasets to compare with hierarchical SOA approaches. **** After feedback : I thank the authors for the clarifications. I understand that the aim of the paper is not to be competitive with other clustering algorithms but my remark on evaluation was essentially out of curiosity. Thanks for the effort to include more details about the implementation.

Strengths: The idea of continuous representation of trees in hyperbolic space is not new but the chosen approach - i.e. direct relaxation of the [12] cost - is very interesting. The proposed formulation and the notion of spread embedding at the core of the method are original. The proof of the correctness of the cost approximation is also very interesting.

Weaknesses: The core weakness of the paper is its readability. The paper is very dense and difficult to read. The reading of the appendix is mandatory to understand the paper. For instance, the core notion of spread embedding is presented in a few lines in section 4.1 and never used afterthat. It seems that the authors have written a long version of this paper and cut it to fit in the required format but without smoothing the core text. The second weakness concerns the reproductibility: no details are given regarding the implementation of the algorithms and the difficulty of the optimization. Notoriously optimizing embeddings in hyperbolic spaces is difficult and often requires tricks to achieve good performances. The authors state that the code will be availble but a paragraph on this subject will be really appreciated (maybe I am wrong and the optimization is straightforward ?). Finally, the experiments show interesting results regarding the capacities of the approach to achieve a good cost in the sense of [12] but I think that for the wider audience results like Table 2 (in appendix) showing purity scores are more interesting. However those results concern only 2 of the dataset, it would be very interesting to have complete results.

Correctness: The reported results are not the average of the runs but the best score, the authors explain clearly and satisfactorily the reasons.

Clarity: See weaknesses. The paper should be reorganized in order to be fully understable without the reading of the annex.

Relation to Prior Work: The details are provided in the Annex. The core article is missing crucial comparison to prior work.

Reproducibility: No

Additional Feedback: The work is very interesting and deserves to be published, but the present form is really hard to read.

[Author Response · NeurIPS 2020]

We sincerely thank the reviewers for their time, feedback, and thoughtful suggestions. The reviewers appreciate the technical novelty in our approach and its theoretical guarantees, and find our work relevant to the NeurIPS community. Reviewer 2 (R2) mainly asked us to add more baselines and experiments, while Reviewer 3 (R3) and Reviewer 4 (R4) asked us to improve our presentation. We respond in more detail below, and took all comments into account in our revised version.

**Evaluation (R2, R4)**  R2 and R4 suggested to include other Hierarchical Clustering (HC) evaluation metrics. We would like to first clarify the claims and evaluation of our work. Our primary contribution is the development of a novel technical approach to optimize over discrete trees, by showing an equivalence between trees and constrained hyperbolic embeddings. Our approach can be used to apply machine learning techniques toward solving any combinatorial search problem involving trees. Thus, our goal is not to show an advantage on different heuristics, but rather to optimize a single well-defined search problem to the best of our abilities. In the context of HC, we focus on Dasgupta's cost (DC), which is a well-studied objective with known guarantees when there is an underlying ground-truth hierarchy [16] (such results have not been established for metrics like dendrogram purity (DP) [24]). We made this clear in our updated draft and included DP scores in the Appendix for completeness, observing that this metric does not correlate well with DC on some datasets.

**Approximation Ratio (R3)**  R3's main concerns are two clarifications about our approximation results. The first asks if the approximation result (Thm 4.1) only holds for the optimal embedding. The answer is no: we prove the stronger result (Lemma C.1) which gives the $(1+\epsilon)$ approximation result when decoding *any* spread embedding, demonstrating the generality of this novel technique. R3's second concern is about the fact that we don't provide an approximation for the continuous optimum. It is known that no constant factor approximation is possible; otherwise, our decoding result would refute known hardness results (under Small-Set Expansion, see [11]). Achieving better (e.g. polylogarithmic) approximations, is an interesting future direction for this work, but currently out of reach, as optimizing non-convex objectives is a recognized difficult problem in the community. We thank R3 for raising these questions and have clarified these in our updated draft.

**HC Baselines (R2)**  We thank R2 for the suggestions to improve our experiments. First, we clarify that the application considered in this work is standard *similarity*-based HC, where the input is only pairwise similarities, rather than features representing the datapoints. This is a well-established setting for analyzing the theoretical guarantees of HC algorithms [16, 33], and also the setting studied by Dasgupta [19]. This setting rules out the Hierarchical K-Means (HKM) (Q2) and decoding (Q3) baselines suggested by R2 as both methods require features as input. In our work, we only compared to methods that have access to the same input information as HYPHC (i.e. similarities), such as Bisecting K-Means, a top-down method which is the direct analog of HKM in a similarity-based context [33]. To address this ambiguity for future readers, we clearly framed the problem setup in our updated draft. To answer R2's question (Q3), we ran the HYPHC decoding without learning embeddings; as expected by R2, this method does significantly worse than HYPHC since input features are not hyperbolic (e.g. 3.411 and 3.288 DC for Zoo and Spambase respectively, versus 2.802 and 3.126 for HYPHC).

**End-to-end Task (R2)**  R2 requested clarification about how the test data was used in the auxiliary end-to-end task. We followed a standard graph-based semi-supervised learning setting, where we have all nodes but only train labels at train time. Note that the purpose of this auxiliary experiment is a simple showcase of the benefits of joint training, which our approach make possible due to its continuous formulation; this can be easily extended to other learning scenarios (e.g. by adjusting embeddings as new examples are provided). We clarified this point in our paper. We also thank R2 for suggesting to measure the performance of a simple classifier that does not perform any clustering step (Q1). Example of such classifiers in a graph context include the Label Propagation (LP) algorithm and we added LP numbers in our updated draft. We find that LP is outperformed by our approach on all datasets (e.g. 76.7 and 46.8 accuracy for LP versus 84.4 and 50.6 for HYPHC on iris and glass respectively), suggesting that clustering learns meaningful partitions of the input similarity graph.

**Presentation (R2, R3, R4)**  We thank the reviewers for their specific comments which helped us re-organize our paper to improve its readability. We included examples and explanations from the Appendix into the main body, such as intuition about new technical concepts (e.g. spread embeddings) (R4), details about the hyperbolic LCA computations (R2) and a more detailed related work (R4). We also clarified the mapping from points to embeddings (R3), which is an embedding lookup (line 154).

**Reproducibility (R4)**  R4 requested more details about the optimization of hyperbolic embeddings. Thanks to the development of Riemannian optimization softwares (e.g. geoopt, 2020), the optimization of hyperbolic embeddings was straightforward, without requiring tricks such as clipping. The hyper-parameters used were learning rate, temperature, batch size, and number of triplets. We added a detailed paragraph describing our implementation and will make it publicly available.



[Meta-Review · NeurIPS 2020]

Throughout discussion among reviewers with the author response, all reviewers agree with the novelty and the significance of the theoretical contribution of this paper, which provides approximation guarantees of the proposed embedding. While reviewers raised a concern about empirical performance regarding with computational cost and parameter tuning, they are common problems for other clustering approaches and are not crucial problems of the proposal. Hence I recommend acceptance of this paper.